# Topological barrier to Cas12a activation by circular DNA nanostructures facilitates autocatalysis and transforms DNA/RNA sensing

Fei Deng [1,2,8], Yi Li [1,2,8] ✉, Biyao Yang[1,2], Rui Sang [1,2], Wei Deng[3], Maya Kansara[4,5,6], Frank Lin [4,7], Subotheni Thavaneswaran[4,5,7], David M. Thomas [4,5,6] & Ewa M. Goldys [1,2]

Control of CRISPR/Cas12a *trans*-cleavage is crucial for biosensor development. Here, we show that small circular DNA nanostructures which partially match guide RNA sequences only minimally activate Cas12a ribonucleoproteins. However, linearizing these structures restores activation. Building on this finding, an Autocatalytic Cas12a Circular DNA Amplification Reaction (Auto-CAR) system is established which allows a single nucleic acid target to activate multiple ribonucleoproteins, and greatly increases the achievable reporter cleavage rates per target. A rate-equation-based model explains the observed near-exponential rate trends. Autocatalysis is also sustained with DNA nanostructures modified with fluorophore-quencher pairs achieving 1 aM level (<1 copy/µL) DNA detection ($10^6$ times improvement), without additional amplification, within 15 min, at room temperature. The detection range is tuneable, spanning 3 to 11 orders of magnitude. We demonstrate 1 aM level detection of SNP mutations in circulating tumor DNA from blood plasma, genomic DNA (*H. Pylori*) and RNA (SARS-CoV-2) without reverse transcription as well as colorimetric lateral flow tests of cancer mutations with ~100 aM sensitivity.

The unprecedented discoveries of CRISPR/Cas systems continue to transform growing areas of biomedicine, synthetic biology, and biotechnology[1–4]. Integrated with guiding RNA molecules (gRNA, crRNA or sgRNA[5]), programmable endonucleases from Class 2 CRISPR/ Cas systems offer unique tools for gene editing and beyond[6–9]. Among those, Cas12a ribonucleoprotein (RNP), with its special characteristics of sequence-dependent *cis*-cleavage and sequence-independent *trans*-cleavage upon RNP activation by a nucleic acid target[10], offers a

breakthrough approach to molecular diagnostics. Low-cost, user-friendly CRISPR/Cas-based assays for nucleic acid detection targeting specific sequences with single nucleotide selectivity[11] can now be developed with ease by gRNA replacement[9,12,13]. These assays employ the *trans*-cleavage enzymatic activity of Cas12a RNP to catalyze degradation of single-strand DNA reporters. Its limited catalytic speed for DNA cleavage (with reported values from 0.07 to 7 turnovers per sec[14]) is insufficient for clinically relevant (<pM) DNA detection without

[1]Graduate School of Biomedical Engineering, Faculty of Engineering, University of New South Wales, Sydney, NSW 2052, Australia. [2]ARC Centre of Excellence for Nanoscale Biophotonics, University of New South Wales, Sydney, NSW 2052, Australia. [3]School of Biomedical Engineering, University of Technology Sydney, Sydney, NSW 2007, Australia. [4]Garvan Institute of Medical Research, Darlinghurst, Sydney, NSW 2011, Australia. [5]St Vincent's Clinical School, University of New South Wales, Sydney, NSW 2011, Australia. [6]Omico, Australian Genomic Cancer Medicine Centre, University of New South Wales, Sydney, NSW 2052, Australia. [7]NHMRC Clinical Trials Centre, University of Sydney, Sydney, NSW, Australia. [8]These authors contributed equally: Fei Deng, Yi Li. ✉e-mail: yi.li6@unsw.edu.au; lca629@foxmail.com

additional amplification strategies, despite various efforts[15–17], while about a 10-fold lower Cas12a *trans*-cleavage rate for RNA targets[18] also prevents direct RNA detection. Disrupting the one-to-one correspondence between target molecules and activated RNPs in standard CRISPR sensors[10,19] is an efficient way to overcome these limitations.

Autocatalytic reactions in the life sciences and biotechnology, particularly those based on nucleic acids, are common[20–23]. Designed nucleic acid sequences can form DNA circuits[22], or realize alternative (e.g., DNAzyme-induced) autocatalytic feedback loops[24–26]. A recent exploration utilized such an autocatalytic nucleic acid circuit in a CRISPR/Cas biosensing system, which increased the DNA detection sensitivity to the attomolar range without amplification[20]. This design employs the nucleic acid strand replacement scheme to close the circuits[20], while the release of free RNA molecules exposes them to self-degradation risk by activated Cas12a RNPs[18]. However, the interactions of designed DNA nanostructures with CRISPR/Cas12a endonuclease activity remain unexplored, and their autocatalytic potential has not yet been identified.

In this work, we report a hitherto undiscovered property of the Cas12a RNPs - highly selective restriction of *trans*-cleavage activation by a specially designed small circular DNA nanostructure (referred to as Cir-mediator) which partially matches the gRNA sequence. The *trans*-cleavage can be restored by other activated Cas12a RNPs that cleave and linearize this Cir-mediator. Building on this finding we establish the <u>A</u>utocatalytic <u>Cas12a</u> <u>C</u>ircular DNA <u>A</u>mplification <u>R</u>eaction (AutoCAR) system, where the standard "one target - one Cas12a RNP" activation pattern[10,19] is broken, allowing a feedback loop for Cas12a RNP activation to generate *trans*-cleavage products at very high rates. This increased output of *trans*-cleavage products is attributed to the increased number of Cas12a RNPs activated by our DNA nanostructures, and their non-linear (near-exponential) trend confirms the presence of autocatalysis. A rate equation-based model of the AutoCAR system is formulated to interpret the observed *trans*-cleavage rates. The AutoCAR system is applied for nucleic acid diagnostics where 1 aM sensitivity (-1 copy/μL) for DNA is achieved without any additional signal amplification at room temperature. A similar sensitivity (-1.4 copies/μL) is achieved for the detection of genomic DNA from a pathogen, *H. pylori*. Furthermore, as autocatalysis overcomes the limitation of low Cas12a *trans*-cleavage rates for RNA targets, we are able to demonstrate RNA detection with Cas12a RNPs alone, without reverse transcription and additional amplification schemes, by identifying 0.36 copies/μL (-1 aM) of SARS-CoV-2 genomic RNA. In

addition, we show that Cir-mediator conjugates with a fluorophore and a matching quencher can simultaneously establish an autocatalytic reaction with Cas12a RNPs and act as highly efficient reporters (Cir-reporters). Nucleic acid sensing with Cir-reporters requires only one other ingredient, Cas12a RNPs, in precise correspondence to classical CRISPR sensors - but sensing performance is dramatically improved in terms of sensitivity, reaction speed and detection range, because signal amplification is theoretically unlimited. As a result, we show that - without any additional amplification reaction or device - it is possible to detect DNA at 1 aM level (<1 copy/μL) sensitivity under 15 min at room temperature, and with uncompromised specificity compared to standard CRISPR/Cas sensors. This is demonstrated by detecting an oncogenic mutation in blood plasma from an animal model of colorectal cancer. We further confirm the feasibility of detecting the same mutation in patients' plasma samples, opening the pathway to affordable genetic tests for circulating tumor DNA in the clinic. Finally, we modify of Cir-mediators to create colorimetric reporters (Cir-color tags) which are simultaneously able to establish an autocatalytic reaction network with Cas12a RNPs and demonstrate a lateral flow assay for a cancer mutation with attomolar level sensitivity. These results expand the boundaries of current CRISPR/Cas biotechnology and pave the way for significant clinical applications and beyond.

## Results

### Introducing the AutoCAR autocatalytic reaction scheme

In a standard CRISPR/Cas12a activation scheme[19,27], the Cas12a endonuclease and gRNA form a functional RNP which then binds nucleic acid targets with complementary sequence to the gRNA spacer region adjacent to an A/T-rich PAM region. Once bound to a target DNA, the Cas12a RNP forms a double strand structure (R-loop) between complementary sequences[27,28]. The Cas12a RNP thus activated is capable of *trans*-cleavage which indiscriminately degrades single-strand DNA (ssDNA) sequences (Fig. 1A). This endonuclease activity is typically evaluated by the cleavage and subsequent unquenching of ssDNA-linked quenched fluorescent reporters and it may be used to amplify the nucleic acid detection signal[7,9].

Our AutoCAR CRISPR/Cas12a activation scheme employs a specially designed small DNA nanostructure termed "Cir-mediator" comprising a double strand (dsDNA) and a ssDNA region (Fig. 1B). The dsDNA region contains both a A/T-rich PAM sequence and a sequence matching the gRNA spacer sequence of Cas12a RNP. When the ssDNA region of the Cir-mediator is broken, the nanostructure changes its

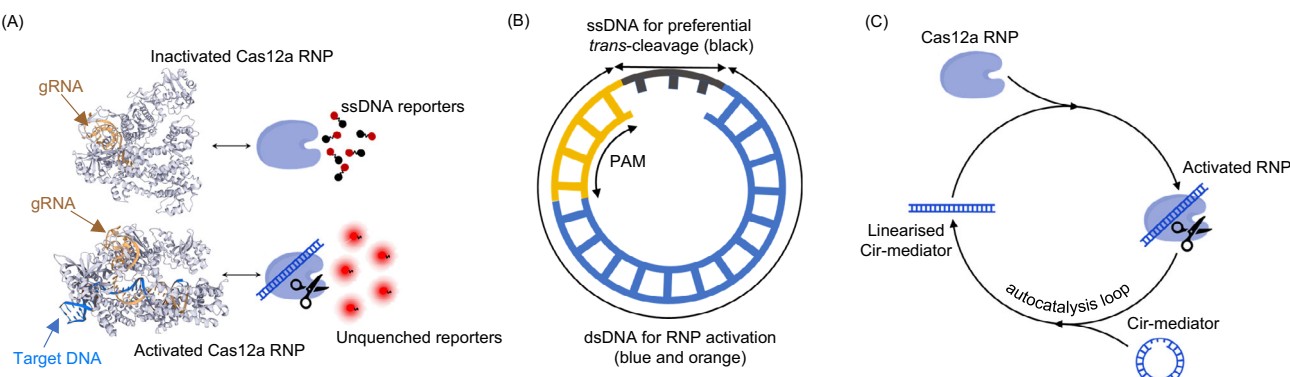

**Fig. 1 | Introducing the autocatalysis loop in the AutoCAR system. A** Standard CRISPR/Cas12a activation scheme. The binding of a nucleic acid target induces activation of Cas12a *trans*-cleavage activity which can be revealed by the cleavage and unquenching of fluorescent quenched reporters[11] (PDB 3D Protein Feature View, 5XUS = activated Cas12a RNP, 6NME = inactivated Cas12a RNP). **B** The design of Cir-mediator. This small circular DNA nanostructure (-20 nt) combines a ssDNA (black) and a dsDNA sequence. The dsDNA incorporates a PAM region (orange) and

a target region (blue) which is complementary to the guide RNA in Cas12a RNP (Ribonucleoprotein). **C** Reaction schematics for Cir-mediator-induced CRISPR/Cas12a autocatalysis reaction. The ssDNA sequence in the Cir-mediator is preferentially cleaved by the activated Cas12a RNP. The activated RNP efficiently cleaves ssDNA but not the dsDNA region of Cir-mediator. This linearizes the Cir-mediator molecule and exposes its dsDNA region to a new inactive RNP, thus closing the loop of the autocatalytic reaction.

topology adopting a linear shape. We unexpectedly found that for specific lengths, the original circular Cir-mediator structure and its linearized form produce significantly different Cas12a activation patterns. Hence, we hypothesized that upon a suitable initiation of the reaction (by at least one linear DNA molecule complementary to Cas12a gRNA), such Cir-mediators and their matching Cas 12a RNPs can set up an autocatalytic reaction network (Fig. 1C). Autocatalysis takes place because activated Cas12a RNPs preferentially cut the short ssDNA region of Cir-mediators, which transforms the Cir-mediator back to a linearized form. These newly formed linear dsDNA oligos matching the gRNA of the RNPs cause activation of further RNPs thus closing the autocatalysis loop of the AutoCAR system.

## Small circular DNA targets induce negligible Cas12a activation

We explored the interaction of nucleic acid targets with Cas12a RNPs in linear ssDNA (L-ssDNA) structures of varying length, complementary to gRNA in the Cas12a RNP and with an A/T-rich PAM region (Fig. 2A)[9,11]. The UNAFold simulations[29] ruled out complex secondary structures in our conditions (Supplementary Fig. 1). As previously reported[10], with the total length of the target ssDNA region decreased from 27 nt to 15 nt, and the binding length of DNA to gRNA (blue region in Fig. 2A) decreased from 21 nt to 9 nt, the levels of Cas12a RNP trans-cleavage measured by fluorescence of ssDNA-linked fluorescent quenched reporters also decreased by over 90% (Fig. 2B). Similar but less pronounced Cas12a RNP activation pattern was also found for varying

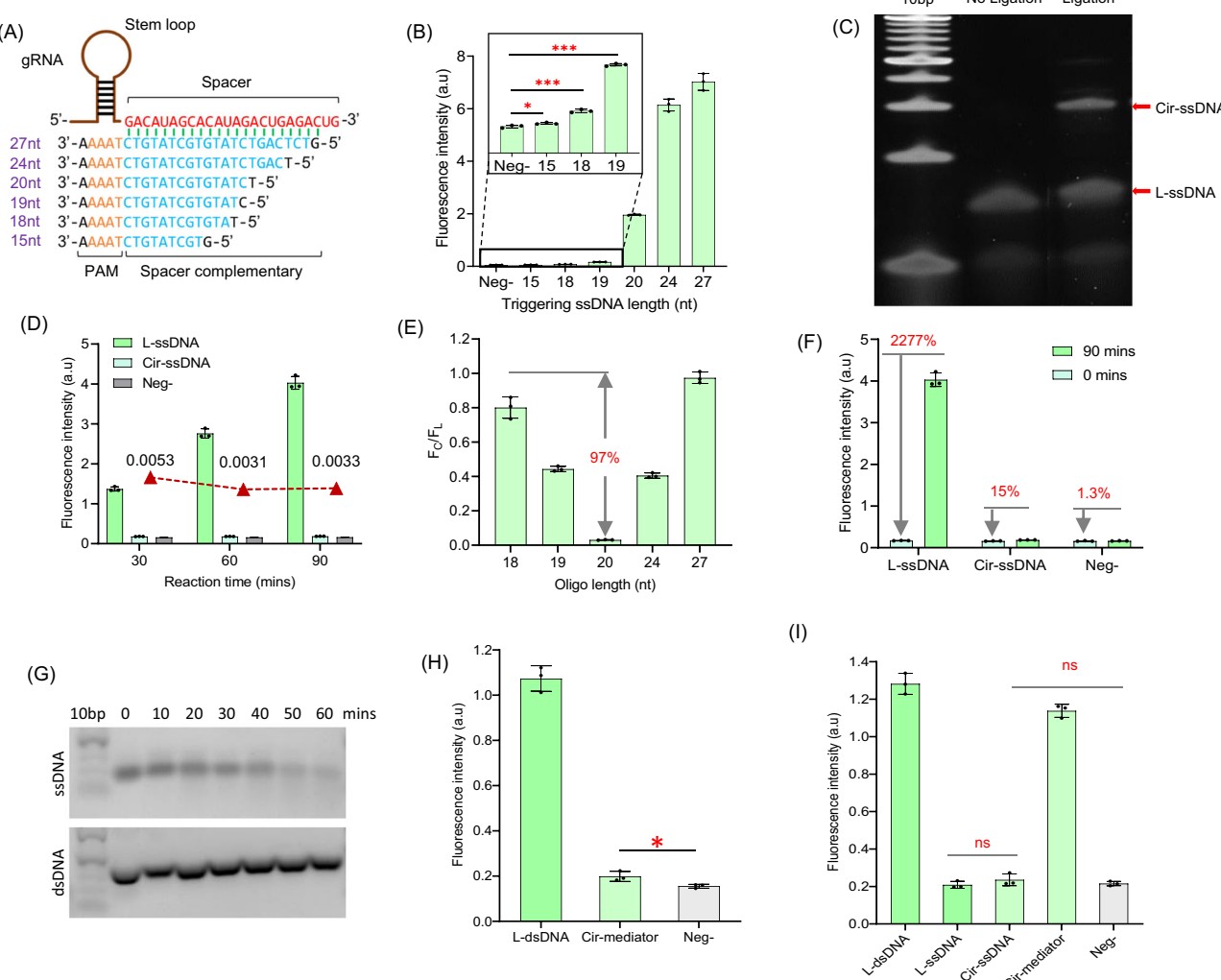

**Fig. 2 | Small circular ssDNA nanostructures produce diminished Cas12a activation. A** Schematic design of L-ssDNA (Linear ssDNA) oligos used to activate Cas12a trans-cleavage. Their total length ranges from 15 nt to 27 nt which include a PAM (orange) and a gRNA binding region (blue). **B** Fluorescence signal produced by Cas12a RNPs triggered by L-ssDNA oligos of different lengths ($P_{(Neg-15)} = 0.0379$, $P_{(Neg-18)} = 0.0002$, $P_{(Neg-19)} < 0.0001$, $n = 3$ independent reactions). The oligos with total length of 15 nt, 18 nt, and 19 nt show less than 15% of the signal induced by 20 nt ssDNA. (Method 2). **C** PAGE-gel electrophoresis verifies the formation of Cir-ssDNA nanostructure made from L-ssDNA. (10 bp DNA marker applied, Method 3). **D** The change of Cas12a RNP trans-cleavage activation patterns for different DNA nanostructures ($n = 3$ independent reactions). The Cas12a RNP activation pattern for 20 nt Cir-ssDNA shows low level of trans-cleavage activity ($P < 0.05$, red dots) which is negligible compared to its linear form (L-ssDNA). (Method 2). **E** The Cas12a RNP activation efficiency ratios between Cir-ssDNA and L-ssDNA of different lengths. $F_C$ and $F_L$ represent the fluorescence signal generated by Cir-ssDNA- and L-

ssDNA-activated Cas12a trans-cleavage ($n = 3$ independent reactions). The most pronounced activation efficiency differences between Cir-ssDNA and its linear form are found for 20 nt ssDNA. (Method 2). **F** Comparison of differences in the trans-cleavage intensity for 20 nt long L-ssDNA and Cir-ssDNA ($n = 3$ independent reactions). The trans-cleavage activity of the Cir-ssDNA molecules is much lower compared to its linear conformation. (Method 2). **G** Gel electrophoresis demonstrates that trans-cleavage of Cas12a efficiently degrades ssDNA but not dsDNA. (10 bp DNA marker applied. Method 4). **H** The Cas12a activation by Cir-mediator DNA nanostructure is very low compared to its corresponding L-dsDNA (Linear dsDNA) conformation ($P = 0.0324$, $n = 3$ independent reactions). (Method 2). **I** Trans-cleavage treated DNA molecules (prepared from 20 nt L-ssDNA oligos) are effective as target DNA for Cas12a RNP activation ($n = 3$ independent reactions). (Method 5). (Two tails student T test, *$P < 0.05$, **$P < 0.005$, ***$P < 0.001$, ns = not significant, Error bars represent mean ± SD, a.u = arbitrary units).

lengths of dsDNA targets (Supplementary Figs. 2, 3). We then discovered that forming a circular ssDNA structure (Cir-ssDNA) from the target ssDNA (Fig. 2C, Supplementary Figs. 4, 5, 36) strongly suppresses Cas12a RNP activation, even though the L-ssDNA from which it was formed produces effective activation (Fig. 2D, Supplementary Fig. 6). The lowest ratio of activation efficiencies between Cir-ssDNA and linear ssDNA was found for 20 nt long ssDNA, with only about 3% of Cas12a activation remaining when circular ssDNA nanostructures were used as a target (Fig. 2E). Although this 20 nt long Cir-ssDNA still yields a statistically higher fluorescence signal compared with negative control (~15% higher, Fig. 2F), it is much lower (~23 times lower) compared with L-ssDNA of the same length (in a 90 min reaction, Fig. 2F). Extending Cir-ssDNA from 20 nt to 27 nt progressively restores RNP activation, while Cir-ssDNA that is shorter than 20 nt produces a considerably lower level of overall Cas12a activation even in a linear form (Supplementary Fig. 7). Therefore, 20 nt oligos (L-ssDNA and Cir-ssDNA) were selected for further investigations. To control *trans*-cleavage activity by using such circular DNA structures, a complementary DNA (cDNA) oligonucleotide was bound to the 20 nt Cir-ssDNA covering the PAM and gRNA-binding regions of the Cir-ssDNA forming a Cir-mediator (Method 1, Supplementary Fig. 8). Informed by reports that *trans*-cleavage of Cas12a effectively degrades ssDNA backbones but not dsDNA[10,19], we verified by gel electrophoresis that the Cir-mediators or its dsDNA region survives the exposure to Cas12a *trans*-cleavage. However, the same L-ssDNA sequence was significantly degraded after 60 min of *trans*-cleavage digestion and failed to activate its corresponding Cas12a RNPs (Fig. 2G, Supplementary Figs. 9–11). Finally, we characterized the Cas12a activation pattern of the Cir-mediators where we found a similar restriction of Cas12a activation by Cir-mediators as by Cir-ssDNA (Fig. 2H, Supplementary Fig. 12).

## Linear residues of Cir-mediators activate Cas12a RNP *trans*-cleavage

We investigated whether Cas12a RNP activation is restored by linearization of the Cir-mediators. To this aim, the Cir-mediators were exposed to pre-activated Cas12a RNPs (expected to cut their single strand region), and then the products were allowed to interact with corresponding Cas12a RNPs (Method 5). The results show that the Cir-mediator (or its L-dsDNA form) was able to effectively activate its corresponding Cas12a RNPs, while the ssDNA molecules (Cir-ssDNA or

its L-ssDNA form) similarly exposed to *trans*-cleavage yielded a significantly reduced Cas12a RNP activation efficiency (Fig. 2I, Supplementary Fig. 13). In addition, with increasing concentration of *trans*-cleavage-treated Cir-mediators, the downstream Cas12a activation efficiency was also increased (Supplementary Fig. 14). Combined, these data indicate that Cas12a *trans*-cleavage can break the ssDNA region of the Cir-mediator structure, thus returning its dsDNA region to linearity, which then can serve as a new target for activation of further Cas12a RNPs.

## Mechanism of restricted Cas12a activation by circular DNA structure

To further investigate the mechanism of the observed restricted activation during the Cir-mediator - Cas12a RNPs interaction, fluorescence resonance energy transfer (FRET) assays were carried out, with the donor (D) Cy3 fluorophore at the 5' end of gRNA, and an acceptor (A) Cy5 fluorophore on an internal dT nucleotide adjacent to the PAM region of the 20 nt target strand (Fig. 3A). The sequence location of the FRET acceptor in linear (L-ssDNA or L-dsDNA) and circular (Cir-ssDNA or Cir-mediator) molecules bound to gRNA is the same, due to basic Watson-Crick complementarity. In comparison to the original linear ssDNA which is known to stably bind to the corresponding Cas12a RNP, both Cir-ssDNA and Cir-mediators showed a significant (about 72%) decrease in the FRET signal which was only slightly higher (6 %) than the background (no Cas RNP, only oligos, Fig. 3B, Supplementary Fig. 15). Since the overall reaction environment of Cas12a RNP tested with linear and circularized DNA was the same (Method 2), we considered whether the FRET signal change could be attributed to a distance change between donor and acceptors due to the binding of oligos to the RNP. The extent of this distance change is limited by the distance change taking place upon RNP activation. Molecular simulations by using PDB 3D Protein Feature View[30] were then carried out to estimate the donor-acceptor distance in inactivated (3.564 nm) and activated (3.732 nm) Cas12a RNP (Fig. 3C). Their difference of 0.168 nm is due to Cas12a RNP configuration change which takes place during RNP activation. The FRET efficiency in an activated Cas12a RNP observed here was 86% (Supplementary Fig. 16) corresponding to donor-acceptor distance of 3.99 nm, according to the relationship $E_{FRET} = 1/1 + \left(\frac{R}{R_0}\right)^6$, where $R_0$ is the Forster distance for Cy3-Cy5 of 5.4 nm and $R$ is Cy3-Cy5 distance[17]. The 0.168 nm distance change of Cas12a protein configuration produces only 0.03% FRET efficiency

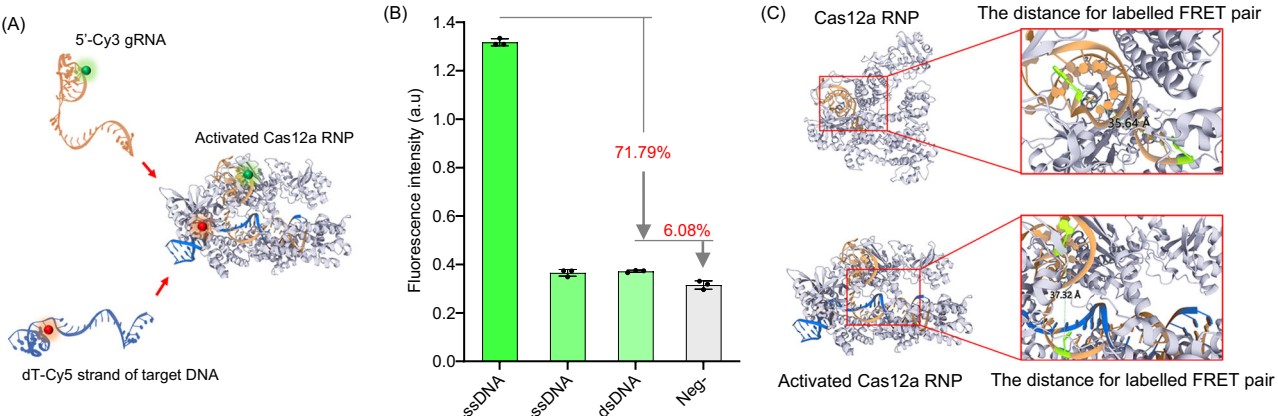

**Fig. 3 | Different interaction of circular and linear conformations of DNA oligos with Cas12a RNP. A** Schematics of FRET assays to investigate the binding interaction of Cas12a RNP and its target DNA. **B** Compared to the original linear ssDNA conformation (L-ssDNA), the FRET signal of circularized ssDNA (Cir-ssDNA) is significantly lower, and the addition of cDNA to form the Cir-mediator structure results in a 71.79% decrease of the FRET signal, which is only 6.08% higher than the

background signal (*n* = 3 independent reactions, Methods 6, S5). **C** Molecular simulation estimates of distance changes between FRET donor and acceptor due to the Cas12a conformation changes before and after activation (PDB 3D Protein Feature View, 5XUS = activated Cas12a RNP, 6NME = inactivated Cas12a RNP). Donor-acceptor distance in inactive RNP is 3.56 nm and in activated RNP it is 3.73 nm. (Error bars represent mean ± SD, a.u = arbitrary units).

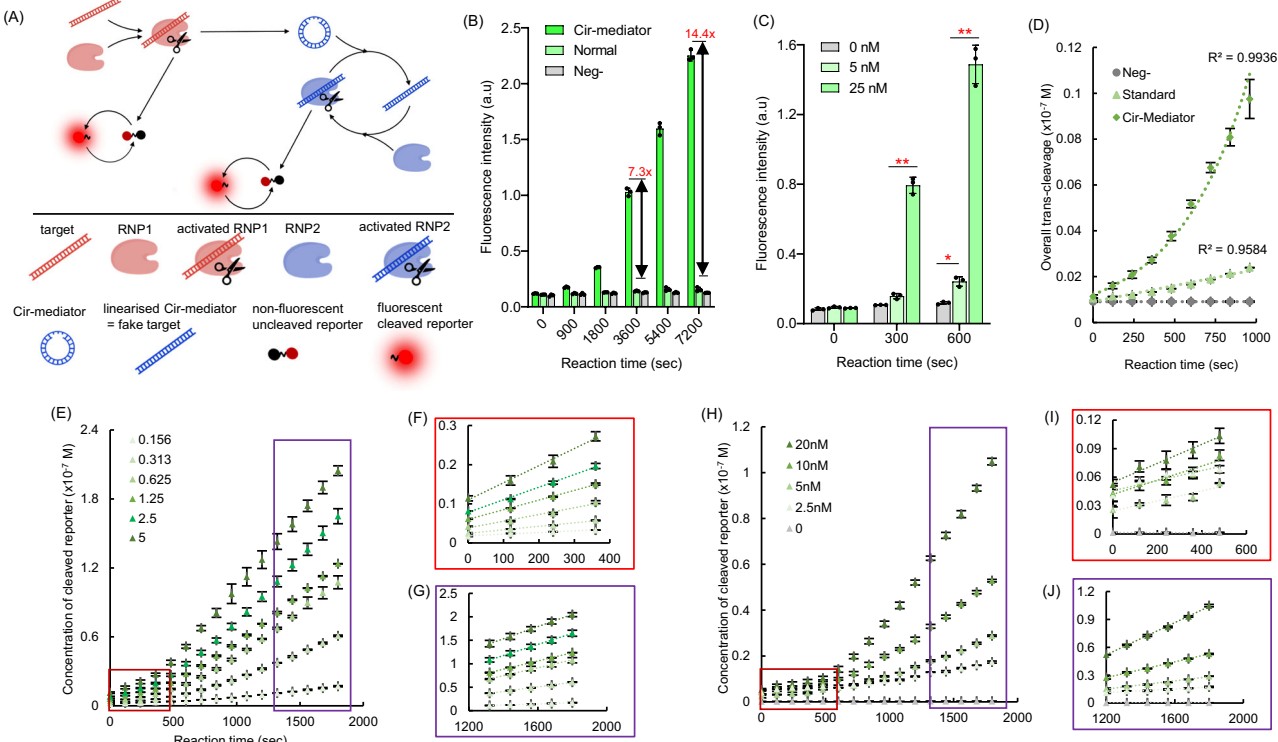

**Fig. 4 | Cir-mediator-induced Cas12a autocatalysis amplifies *trans*-cleavage in AutoCAR-1 scheme. A** The AutoCAR-1 scheme for amplified ssDNA cleavage using ssDNA-linked fluorescence-quenched reporters. **B** The fluorescence signal intensity differences between reporter *trans*-cleavage in a standard CRISPR/Cas12a and the AutoCAR-1 systems measured by the fluorescent signal produced by cleavage of fluorescent quenched ssDNA reporters (*n* = 3 independent reactions). In comparison with a standard Cas12a catalytic system without Cir-mediators, the reporter *trans*-cleavage increased 7.3 times over 3600 s (60 min) and 14.4 times over 7200 s (120 min) reactions at room temperature. (Method 7). **C** The reporter *trans*-cleavage in AutoCAR-1 increased with increasing Cir-mediator concentration ($P_{300(0-5nM)}$ = 0.0046, $P_{600(0-5nM)}$ = 0.0016, $P_{600(5-25nM)}$ < 0.0001, *n* = 3 independent reactions). (Method S6). **D** Time dependence of the fluorescence signal for AutoCAR-1 in comparison to a standard CRISPR/Cas12a reaction (*n* = 3 independent reactions). Our data show differences in reporter *trans*-cleavage kinetics in a standard Cas12a catalytic system (linear trend, $y = 0.014x + 9.1129$, goodness of fit $R^2 = 0.9584$) and Cir-mediator-assisted Cas12a autocatalysis reaction (super-linear trend, exponential fit, $y = 12.102e^{0.0023x}$, goodness of fit $R^2 = 0.9844$). These results indicate that instead of the well-established linear trend[14], AutoCAR-1 produces a nonlinearly increasing signal (Method 7). **E** The influence of different concentrations of reporters in the AutoCAR-1 system with 50 nM Cir-mediators (*n* = 3 independent reactions) (Method 8). **F** Magnified section of Fig. 4E showing the time window 1 (0−6 min). **G** Magnified section of Fig. 4E showing the time window 2 (20−30 min). **H** The influence of different concentrations of Cir-mediators in the AutoCAR-1 system with excess reporters (*n* = 3 independent reactions, Method 9). **I** Magnified section of Fig. 4H showing the time window 1 (0−6 mins). **J** Magnified section of Fig. 4H showing the time window 2 (20−30 min). (Two tails student T test, Error bars represent mean ± SD, *$P < 0.05$, **$P < 0.005$, ***$P < 0.001$, a.u = arbitrary units).

changes, which is much less than the 72% decrease shown in Fig. 3B. Therefore, we conclude that the fluorescence signal changes in Fig. 3B are due to the significantly reduced binding affinity of Cir-mediators to RNPs compared to their linearized form.

## Cir-mediators increase the overall levels of fluorescent reporter cleavage but not Cas12a cleavage rates

Having established that the Cir-mediators can be linearized by RNP *trans*-cleavage, a step which can then induce further Cas12a RNP activation, we set up an <u>Auto</u>catalytic <u>C</u>as12a Circular DNA <u>A</u>mplification <u>R</u>eaction 1 (AutoCAR-1) system (Fig. 4A). In AutoCAR-1, we combined two types of Cas12a RNPs loaded with two different gRNA sequences. Cas12a RNP1 with gRNA1 complementary to the DNA target was used to initiate the reaction. As the original target activates RNP1, the RNP1 *trans*-cleavage cuts the surrounding ssDNA including the ssDNA region of Cir-mediators. The autocatalysis loop is then set up with Cas12a RNP2 whose gRNA2 matches the dsDNA region of the Cir-mediators. Short ssDNA-linked fluorescent quenched reporters[31] monitored the overall reporter *trans*-cleavage rate (Method 7). A significantly higher fluorescence signal was found in the AutoCAR-1 system, compared to a standard CRISPR/Cas12a reaction without Cir-mediators (Fig. 4B) which is indicative of elevated reporter cleavage level (7.3 times higher for 1 h and 14.4 times higher for 2 h reactions, in our conditions). With

increasing concentration of Cir-mediators, the fluorescence signal at the same time points also increased (Fig. 4C, Supplementary Fig. 17). These data indicate higher overall levels of fluorescent reporter cleavage in the AutoCAR-1 system than in a standard CRISPR-Cas reaction with the same amounts of reaction-initiating targets. Unlike the standard CRISPR/Cas12a *trans*-cleavage activation scheme[17] which shows a linear correlation between reaction time and fluorescence intensity (Supplementary Fig. 18), the fluorescent signal in the AutoCAR-1 system is able to increase non-linearly with reaction time (Supplementary Fig. 19). Furthermore, for the same concentrations of target DNA there is a notable difference in the time course of fluorescent reporters' cleavage rate in AutoCAR-1 and the standard CRISPR/Cas12a reaction schemes. The standard reaction showed the well-established linear time dependence, but the AutoCAR-1 scheme produced a super-linear, nearly exponential trend (Fig. 4D).

To eliminate the possibility that Cir-mediator molecules are modifying Cas12a RNP *trans*-cleavage rates, similarly to earlier reports for other molecules[17], a standard Michaelis-Menten analysis was carried out[14]. This is appropriate for quasi-steady state conditions which are approximately satisfied in limited time windows. Empirical estimates for *trans*-cleavage rates were deduced from fluorescence data (Fig. 4E−G), whose trends over such limited time windows (0−6 min = TW 1, 20−30 min = TW 2)

are close to linear. The Michaelis-Menten analysis in these two time windows yielded closely similar estimates for the turnover rates $K_{cat} = 0.08 \pm 0.02\,s^{-1}$ (Supplementary Fig. 20, Supplementary Table 1), which are well aligned with published $K_{cat}$ values for LbCas12a (-0.07 to -0.09 s⁻¹)[14]. This is consistent with Cas12a *trans*-cleavage rates being unaffected by the presence of Cir-mediators, hence pointing to activation of additional Cas12a RNPs as a reason for enhanced reporter cleavage.

### Autocatalysis in the AutoCAR-1 system – theoretical model and comparison with experiment

In order to understand the autocatalytic network in the AutoCAR-1 system, we set up a model of chemical kinetics rate equations expanding the established approach in a standard CRISPR/Cas12a reaction[14] (Supplementary Note). This model introduces an auto-catalytic loop for Cas12a, as well as two separate catalytic loops for the fluorescent reporters (Fig. 4A, Equations 1, 2, 5–12). It allows us to establish, in the Briggs-Haldane approximation, that the observed reporter cleavage rate, in specific reaction conditions, is proportional to the concentration of activated RNPs (Equation 24). This supports the application of the Michaelis-Menten theory in defined time windows to estimate the observed *trans*-cleavage rates (Fig. 4E, Supplementary Fig. 20). The model also allows to estimate the number of activated Cas molecules per single target molecule, $R_{Cas12a/target}$ (Fig. 4H–J, Supplementary Figs. 21, 22). We found that $R_{Cas12a/target}$ in the presence of 20 nM Cir-mediators over the reaction time of ~ 1500 s increased by 3 orders of magnitude (a factor of $1600 \pm 500$, Supplementary Table 2). This result confirms that in AutoCAR-1 a single target molecule leads to the activation of a large number of additional RNPs over the course of the reaction (Supplementary Fig. 23). Overall, the $R_{Cas12a/target}$ showed a positive linear correlation with the concentration of Cir-mediators in the AutoCAR-1 reaction system (Supplementary Fig. 24), which indicates an elevated numbers of activated Cas12a RNPs for same level of target nucleic acid sequences. The reporter cleavage rate trends (fluorescent signal) in various experimental conditions were also revealed, most notably in conditions when none of the reagents becomes depleted in the reaction (Equation 37) when near-exponential growth is predicted, in agreement with results in Fig. 4D and Supplementary Fig. 19. Other experimental conditions, e.g., when Cir-mediators may become depleted but the level of targets is negligible also yield a non-linear reaction rate pattern (Fig. 4H). Finally, once the Cir-mediators (but not reporters) have been depleted, the reaction kinetics returns to the well-known trend of constant rates and fluorescence signals linearly increasing with time[14]. These now reflect the cumulative effect of previous autocatalysis events on the overall reporter cleavage rate, in agreement with Equation 24 which predicts proportionality of the rates to the reporter concentration (Fig. 4E, H, Supplementary Fig. 21). When even the reporters become depleted, the fluorescent signal saturates (Supplementary Fig. 25).

### Cir-mediator-assisted Cas12a autocatalysis (AutoCAR-1) applied to ultra-sensitive DNA and RNA detection

Enhanced, exponentially increased reporter cleavage rates in the AutoCAR-1 system are ideally suited for ultrasensitive nucleic acid detection. We found that AutoCAR-1 was able to reveal the presence of target DNA at 1 aM (Supplementary Fig. 26, Method 7). This is impossible in a standard CRISPR/Cas12a DNA reaction, which only reaches ~ 1–10 pM range without additional amplification strategies (Supplementary Figs. 26, 27, Method 2). In experimental conditions described in Method 10, AutoCAR-1 had a linear range of 3 orders of magnitude from 1 aM to 1 fM (Fig. 5A) and was able to distinguish the changes of target DNA concentrations between 1 to 10 aM (Supplementary Fig. 28). The fluorescence signal of AutoCAR-1 system could also be detected by an alternative readout instrument (a real-time

thermocycler) which achieved the same 1 aM nucleic acid detection sensitivity (Supplementary Fig. 29, Method S9).

In order to assess if AutoCAR-1 can be applied for real pathogen detection, the extracted *H. pylori* whole genome DNA was used as a target to initiate the AutoCAR-1 system with the gRNA sequence of Cas12a RNP1 corresponding to the *H. pylori glm* gene fragment (Method 10). The results confirm that AutoCAR-1 can detect the presence of 1.4 copies/μL (-1 aM range) of *H. pylori* genome DNA in the tested sample at room temperature (Fig. 5B), at comparable sensitivity to the current gold-standard qPCR (Supplementary Fig. 30).

Although Cas12a RNP is, in principle, able to recognize RNA sequences[18], direct RNA detection by Cas12a is impractical due to a very low level of RNA-induced *trans*-cleavage (Supplementary Fig. 31); as a result earlier works relied on the combination of reverse transcription and/or additional amplification strategies. Autocatalysis in our AutoCAR-1 overcomes this limitation, allowing direct RNA detection without reverse transcription or amplification. To demonstrate this, the reporter concentration in AutoCAR-1 was increased (Method 11) and we also added DTT as a Cas12a *trans*-cleavage enhancer (Supplementary Fig. 32, Method 11). In such conditions, AutoCAR-1 detected synthetic RNA, without reverse transcription, at a sensitivity down to 1 aM and with a linear range of 3 orders of magnitude (Fig. 5C). We also detected genomic RNA extracted from SARS-CoV-2 viral particles (Methods 11, S12). AutoCAR-1 was able to detect the presence of the N-gene of SARS-CoV-2 viral genome RNA at the sensitivity of less than 1 copy/μL (360 copies/mL), which is, again, comparable to the current gold standard RT-qPCR (Fig. 5D, Supplementary Fig. 33, Method S13).

### Transforming Cir-mediators into fluorescent reporters suitable for a standard CRISPR/Cas12a biosensing system

To simplify the preparation of the Cir-mediator-assisted autocatalysis system, a magnetic-beads-based click chemistry method was introduced to improve purity of single ring synthesis and increase product concentration (Supplementary Fig. 36). Furthermore, we modified the synthesized Cir-mediators (with the dsDNA sequence identical to the target sequence) by labeling their complementary DNA strand (cDNA) at both ends with a fluorophore and a matching quencher. Their distance is equivalent to the length of the ssDNA linker. These structures, referred to as "Cir-reporters", are linearized by activated Cas12a RNPs producing a fluorescent signal (Fig. 6A). They show a significant difference in fluorescence signals between the circular and linearized conformation (Fig. 6B). The background fluorescence in Cir-reporters is due to incomplete quenching, as seen for example, for ssDNA linker length of 5 nt which is comparable to the FRET distance of the Texas/BHQ2 pair[32]. After cleavage and linearisation, the fluorescence signal significantly increases (by a factor of 16.5) since the fluorophore-quencher distance (18nt, estimated to be 6.12 nm[33]) exceeds the FRET distance of Texas/BHQ2 pair (over 10 nt)[32].

Optimization of fluorescent signal in Cir-reporters was carried out by varying the ssDNA linker length (Fig. 6C) where longer linker lengths increased the background due to increased fluorophore-quencher distance. The Cir-reporters with different linker lengths from 0 to 7 nt were then exposed to activated Cas12a RNPs (Fig. 6D). The fluorescence signal was found to initially increase with linker length, since longer linkers could be more easily cleaved by Cas12 RNP[34], and then it stabilized from 3 nt to 7 nt. Thus, 3nt ssDNA linker length was found to be optimal. The optimized Cir-reporters with 18nt dsDNA and 3nt ssDNA linker were compared, in a standard CRISPR/Cas12a biosensing system, with a commonly used linear ssDNA reporter (TTATT, with identical fluorophore-quencher pairs). The Cir-reporter signal was higher by a factor of 3 (Supplementary Fig. 37), and the limit of detection (LOD) was 10 times lower (0.1 pM) than that of linear ssDNA reporter-assisted CRISPR/Cas12a biosensing system (1 pM) (Fig. 6E).

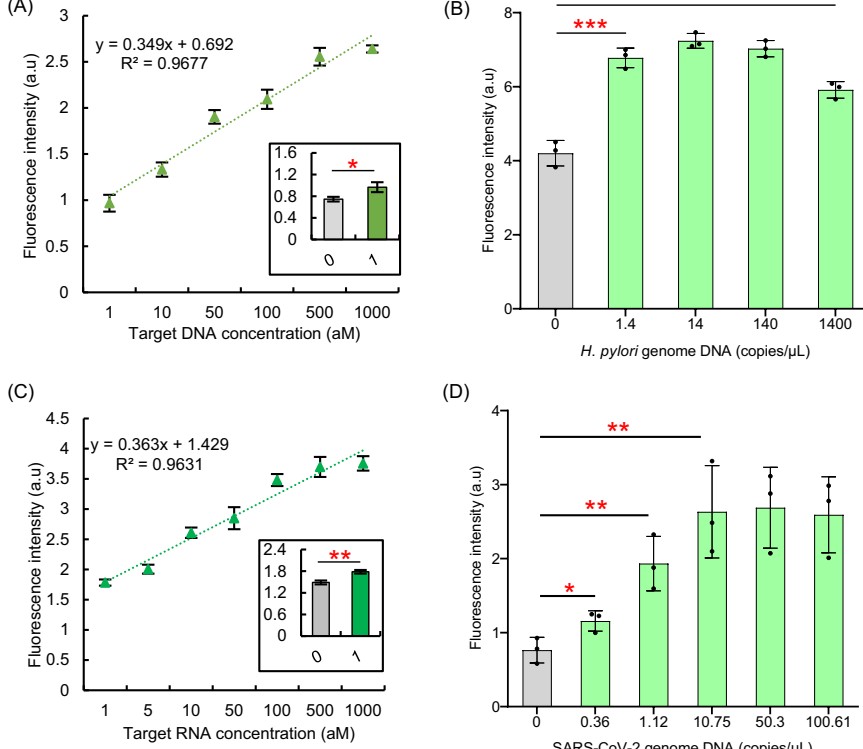

**Fig. 5 | AutoCAR-1 is capable of ultra-sensitive DNA and RNA diagnostics with no amplification and no reverse transcription. A** The calibration curve for AutoCAR-1 DNA detection ($P = 0.01885$, $n = 3$ independent reactions). The system has 3 orders of magnitude linear range, with DNA detection sensitivity down to 1 aM. (Two-tailed $t$ test, Method 7). **B** Detection of *H. pylori* bacterial genome DNA using the AutoCAR-1 system targeting the *glm* gene ($P = 0.0005$, $n = 3$ independent reactions, Method 10). **C** The AutoCAR-1 calibration curve for RNA detection ($P = 0.0023$, $n = 3$ independent reactions). The system investigated here shows 3 orders of magnitude linear range with RNA detection sensitivity of 1 aM. (Two-tailed $t$ test, Methods 11, S10). **D** Detection of the N-gene fragment of SARS-CoV-2 viral genome RNA using the AutoCAR-1 system alone, without reverse transcription

($P_{(0-0.36)} = 0.0368$, $P_{(0-1.12)} = 0.0076$, $P_{(0-10.75)} = 0.0075$, $n = 3$ independent reactions, Method 11). The 1 aM LOD is consistent with exponential growth in the number of Cas12a proteins activated by a single target ~3 orders of magnitude in ~25 min, over approximately twice the duration (~1 h) yields the LOD increase of ~3 orders of magnitude squared, so 6 orders of magnitude lower compared with well-established LOD values for pre-amplification free Cas12a detection systems (1–10 pM). Kinetic fluorescence signal profiles are shown in Supplementary Fig. 34. Due to potential Cas12a off-target effects in genomic nucleic acid materials[56], comparison and screening of different Cas12a targeted sequences can potentially improve the overall background and performance. (Two tails student T test, error bars represent mean ± SD, *$P < 0.05$, **$P < 0.005$, ***$P < 0.001$, a.u = arbitrary units).

## RNP activation efficiency of Cir-reporters and linearized Cir-reporters

We assessed the RNP activation efficiency of Cir-reporters for different lengths of its dsDNA section and the ssDNA linker. As shown in Fig. 7A, a certain amount of RNP activation producing a background signal was observed when the dsDNA length was higher than 18nt, but the background was significantly reduced below 18nt. Since minimal RNP activation by Cir-reporters is desirable, 18nt dsDNA length was selected for further studies[35]. As shown in Fig. 7B, the background signal slightly increased for ssDNA linker lengths between 0 and 3 nt, and then it grew sharply for 5-10 nt, thus the ssDNA linker length below 3 nt was found to be optimal for background control. Additionally, since longer ssDNA linkers are easier to cleave than shorter ones (Supplementary Fig. 38), a 3nt ssDNA linker was selected for the following work[34]. Furthermore, we investigated the RNP activation efficiency by linearized Cir-reporters with 18nt dsDNA and 3nt ssDNA (Fig. 7C). Excellent activation efficiency was found, with fluorescence increases by a factor of over 20, and the signal value was comparable to that from a linear dsDNA conformation (18nt) (Fig. 7D).

## Cir-reporter-assisted autocatalytic sensors (AutoCAR-2 and AutoCAR-3 schemes)

The Cir-reporter-assisted autocatalytic sensor (AutoCAR-2 scheme, Fig. 8A) combines two components, Cas12a RNPs and Cir-reporters, in complete analogy to a standard CRISPR/Cas sensor system. These bifunctional Cir-reporters play a dual role, of a catalytic substrate for *trans*-cleavage by activated Cas RNPs (in analogy to the reporters in a standard CRISPR sensor design[36,37]) and an autocatalytic substrate for the yet-to-be activated Cas12a RNPs. This is because after cleavage they become surrogate targets, due to sequence identity with the real targets. This seemingly minor modification of replacing conventional reporters with Cir-reporters has a profound impact on the reaction network within the sensor, as it sets up autocatalysis. Like in Auto-CAR-1, the target-activated RNP linearizes a number of Cir-reporters which then continue to generate additional activated Cas12a RNPs, and these create a cascade of additional linearized Cir-reporters – each of which reports the detection. This chain reaction continues as long as the Cir-reporter substrate and yet-to-be-activated Cas12a RNPs are not depleted. In comparison with a standard CRISPR/Cas12a sensor whose signal linearly increases with time, the signal observed in AutoCAR-2 grows exponentially (Fig. 8B). The signal increases with increasing supply of Cir-reporters for constant levels of Cas12a RNPs (Fig. 8B), and with increasing concentration of Cas12a RNPs for constant Cir-reporter levels (Fig. 8C). The LOD of 1 aM was achieved, with more than 11 orders of magnitude detection range (Fig. 8D, E). A similar autocatalysis reaction pattern and the same 1 aM LOD was also confirmed by using another Cas12 ortholog, AsCas12a, instead of LbCas12a (Supplementary Fig. 39). This LOD is 6 orders of magnitude lower than in a standard CRISPR/Cas12a biosensing system (1 pM) (Supplementary Fig. 27B). The detection of single nucleotide

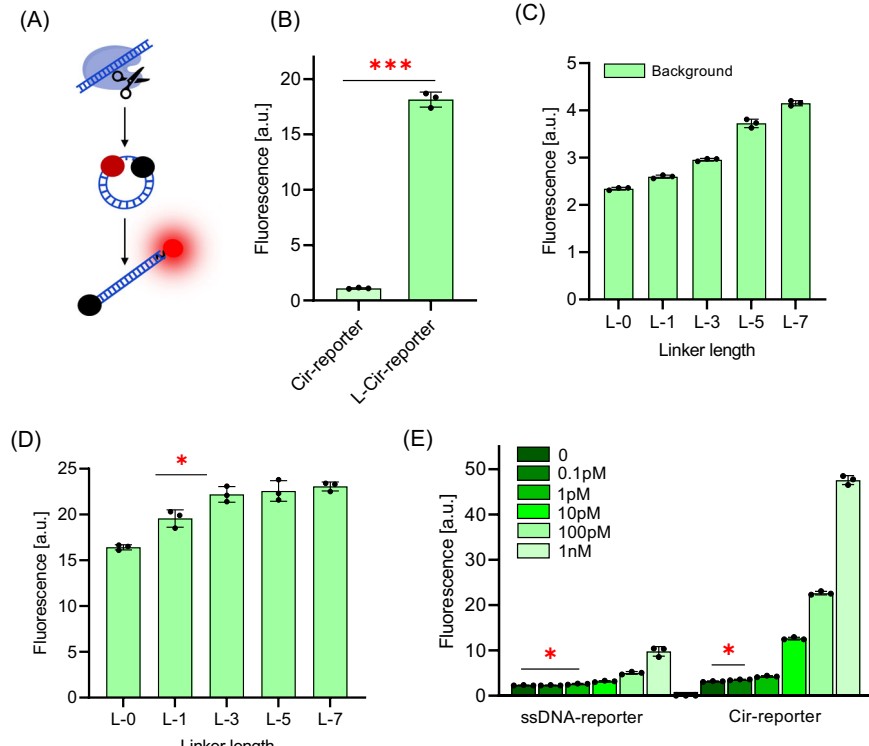

**Fig. 6 | Performance of Cir-reporters in a standard CRISPR/Cas12a biosensing system (Methods 13, 14). A** Unquenching schematics of Cir-reporters; **B** Comparison of the fluorescence signals of a Cir-reporter and linearized Cir-reporter (L-Cir-reporter) (18nt dsDNA with 3nt ssDNA) ($n = 3$ independent reactions); **C** Background signals of Cir-reporters with different linker lengths (18nt dsDNA) ($n = 3$ independent reactions). L-x denotes the linker length of x nt; **D** Investigation of the Cir-reporter linker length in a standard CRISPR/Cas12a biosensing system (18nt dsDNA, 100pM target DNA) ($n = 3$ independent reactions); **E** Comparison of the detection limits of standard CRISPR/Cas12a biosensors with Cir-reporters (18nt dsDNA with 3nt ssDNA) and with linear ssDNA reporters (TTATT) with identical fluorophore-quencher pairs ($n = 3$ independent reactions). (Two tails student T test, Error bars represent mean ± SD, *$P < 0.05$, **$P < 0.005$, ***$P < 0.001$, a.u = arbitrary units).

mismatches in the target sequence has also been demonstrated (Supplementary Fig. 40). The signal in AutoCAR-2 closely follows the predictions of our chemical kinetics rate equations (Supplementary Note) showing an exponential trend (Fig. 7B) provided the yet-to-be activated RNPs and Cir-reporters do not get depleted (Supplementary Fig. 41).

To illustrate versatility, in the AutoCAR-3 system we combined a standard CRISPR/Cas12a system and additional Cir-reporters with corresponding Cas12a RNPs (Supplementary Fig. 42). In this approach, two Cas12a RNPs are used which is similar to AutoCAR-1 (Fig. 4A) but the Cir-mediator and linear ssDNA reporter were replaced by the Cir-reporter. The autocatalysis in AutoCAR-3 serves as a signal amplifier for the standard CRISPR/Cas12a *trans*-cleavage reaction, allowing it to achieve the LOD of 1 aM (Supplementary Fig. 42).

**Application of AutoCAR-3 to quantify circulating tumor DNA (ctDNA) in cancer blood plasma**

To evaluate the feasibility of clinical diagnostics, we confirmed the stability of the synthesized Cir-reporter in clinically relevant human plasma and saliva samples (over 1 h at room temperature, Supplementary Fig. 43). As a proof of concept, we further applied AutoCAR-3 for the detection of ctDNA with an oncogenic mutation (PIK3CA H1047R) in blood plasma of mice with orthotopic human colorectal cancer xenografts (3 animal groups: normal mice, mice bearing human colorectal cancer (CRC-mice), and X-ray treated CRC-mice)[38]. First, AutoCAR-3 was used with a synthetic ctDNA (PIK3CA H1047R) spiked plasma sample (Figs. 9A) and 1 aM sensitivity was achieved in non-diluted plasma, although higher background signal was

observed than in plasma-free controls. The assay range was 4 orders of magnitude as indicated by the AutoCAR-3 calibration curve (Fig. 9B). The AutoCAR-3 system was then used for the detection of ctDNA in 10 μL mice plasma samples. We found that AutoCAR-3 but not the standard CRISPR biosensor was able to distinguish ctDNA from normal and CRC mice (Fig. 9C). Furthermore, lower ctDNA concentration was observed in X-ray treated CRC-mice (Fig. 9C), confirming the feasibility to monitor cancer treatment by using AutoCAR-3. Based on our calibration curve (Fig. 9B), the ctDNA concentrations in CRC and X-ray treatment groups were estimated to be 48.87 aM and 11.6 aM, respectively, while no detectable ctDNA was found in the normal mice group[38].

Finally, we tested the ability of AutoCAR-3 to directly detect PI3KCA H1047R mutations from human plasma. Blood samples were selected from patients with tumors known to harbor PI3KCA mutations, as well as control cancer patients without that mutation, from the Molecular Screening and Therapeutics (MoST) program (ACTRN12616000908437). Plasma samples were assessed for mutation detection using the same PIK3CA H1047R biosensor method (test tube) as for the CRC mice, with a 2-h incubation at ambient temperature. The results show statistically significant discrimination between cancer and control samples (Fig. 9D, Supplementary Fig. 44), which can also serve as proof for the feasibility of DNA single nucleotide polymorphism detection. In addition, we tested if our system can be used for the detection of target DNA in other types of clinical samples such as saliva, by spiking of 1 fM PIK3CA H1047R mutation sequence. The results also show that the presence of targeted PIK3CA H1047R mutation sequence was detected with 100% accuracy (Supplementary Fig. 45).

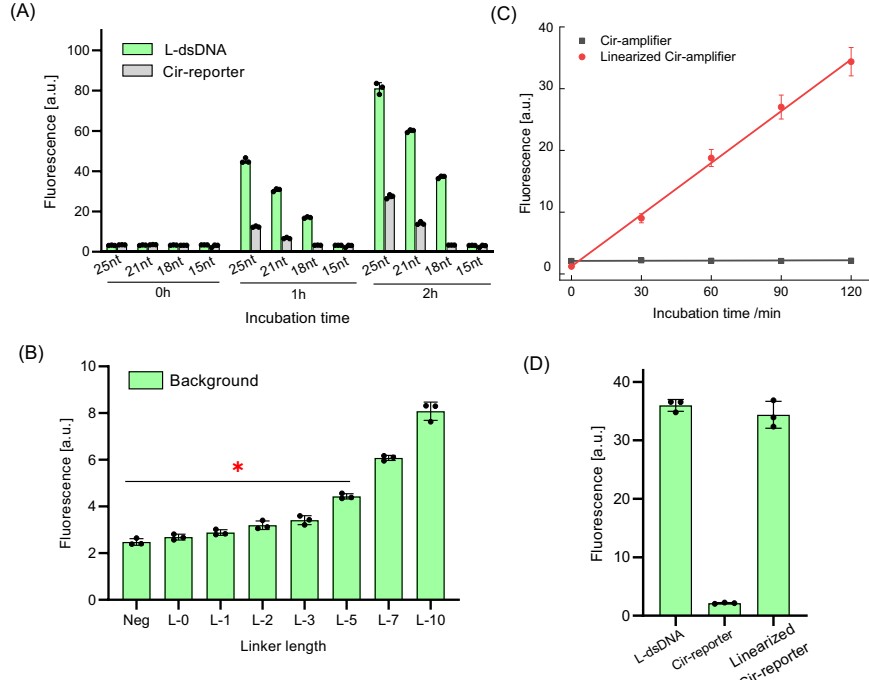

**Fig. 7 | RNP activation efficiency of Cir-reporters and their linearized conformation.** (Methods 15, 16). **A** Evaluation of the dsDNA length in Cir-reporters for the activation of a standard CRISPR/Cas12a biosensing system (with a 3nt ssDNA linker) ($n = 3$ independent reactions). L-dsDNA has been formed by using ssDNA with its c-DNA; **B** Evaluation of the linker length in Cir-reporter for the activation of a standard CRISPR/Cas12a biosensing system (with 18nt dsDNA surrogate target) ($n = 3$ independent reactions). Here L-x denotes the linker length of x nt. Longer linker lengths produce higher background due to reduced FRET effect with more distant quenchers; **C** Evaluation of the RNP activation efficiency of linearized Cir-reporters ($n = 3$ independent reactions); **D** Comparison of the RNP activation efficiency by linearized Cir-reporters and by corresponding linear dsDNA ($n = 3$ independent reactions). (Two tails student T test, Error bars represent mean ± SD, *$P < 0.05$, a.u = arbitrary units).

## Transforming Cir-mediators into colorimetric reporters for lateral flow assays

To demonstrate the suitability of the AutoCAR sensors in point-of-care settings, we converted the Cir-mediator into a colorimetric tag for lateral flow assays (Cir-color tag, Fig. 9E)[39]. The Cir-color tag is closely similar to the Cir-reporter, except for the extension of 18nt cDNA with a 5 nt sequence (CCCCC), addition of biotin on 3' end, and omission of the quencher, while the fluorophore was changed here to FAM. This structure was used for colorimetric readout of an AutoCAR-4 reaction (Fig. 9E) carried out in a test tube (Method 21). When the reaction was deemed to be completed, the mixture was read on a commercial lateral flow (LFA) test strip, where its FAM-containing components conjugated with anti-FAM antibodies (Abs) and gold nanoparticles (Au NPs) from the conjugation pad (Fig. 9F). In the presence of a target in the original reaction the Cir-color tags were cleaved to yield linearized Cir-color tags (Fig. 9E), which were further captured by the secondary antibody on the test line for signal readout due to the loss of their 3'-5C biotin tails (positive test result). Conversely, without a target, the AutoCAR-4 system was not activated; the Cir-color tags remained intact (Fig. 9E) and were captured by the streptavidin on the control line for signal readout and not on the test line (negative test result). This AutoCAR-4 LFA assay was applied to test the plasma ctDNA from normal and CRC-mice (Fig. 9F), and an observable color intensity difference was observed between the CRC-mouse (48.8 aM target) and normal mouse (0 aM target) samples (Fig. 9F).

## Discussion

Setting up reaction networks capable of autocatalysis in CRISPR/Cas12a systems is challenging because these systems have, thus far, been limited to one-to-one pairing of nucleic acid targets and Cas12a RNPs[10,14]. The only viable choice in this scenario is to create a nucleic acid-based autocatalytic circuit accompanied by downstream Cas12a RNP-based detection[20]. However, using circuits that rely on the release of unbound free gRNA as in ref. 20 (e.g., due to reduced double strand binding) presents a risk, because free gRNA is known to be unstable in uncontrolled reaction environments, and it can also be degraded by activated Cas12a *trans*-cleavage[18]. In addition, continuing release of free gRNA has the potential to disrupt the R-loop in activated Cas12a RNP[40], hence leading to the interruption of unspecific ssDNA degradation, which is the key to realize the autocatalysis loop, and may lead to its breakdown. In contrast, in the AutoCAR system demonstrated here, the autocatalysis loop is realized via the interaction between our Cir-moieties (mediators/reporters/color tags) and Cas12a RNPs, where the above considerations do not apply. Once the reaction is initiated by at least a single targeted nucleic acid sequence, the Cir-moieties recruit and activate large amounts of Cas12a RNPs without exposing free gRNA molecules.

The key property of Cas12a RNPs which facilitates autocatalysis is its inability to bind and be-activated by Cir-moieties in their circular topology—which changes when the topology barrier is overcome by *trans*-cleavage, and they become linearized. The process of Cas12a RNP activation requires the unwinding of the double helix structure of the target DNA, which is known to be torsionally regulated[41]. RNA-guided DNA recognition occurs by strand separation of a protospacer target to allow Watson–Crick base pairing between the DNA targeted strand (TS) and the spacer sequence of a gRNA, and the unwinding of a non-targeted strand (NTS)[42]. After *cis*-cleavage, the Cas12a RNP remains bound to the PAM-proximal cleavage product and the RNP undergoes a conformational change enabling *trans*-cleavage. Thus the *trans*-cleavage process is predicated on the formation and dissociation of the R-loop - which requires torque[43]. The Cir-moieties in AutoCAR are very short, corresponding to approximately one or two coils of the

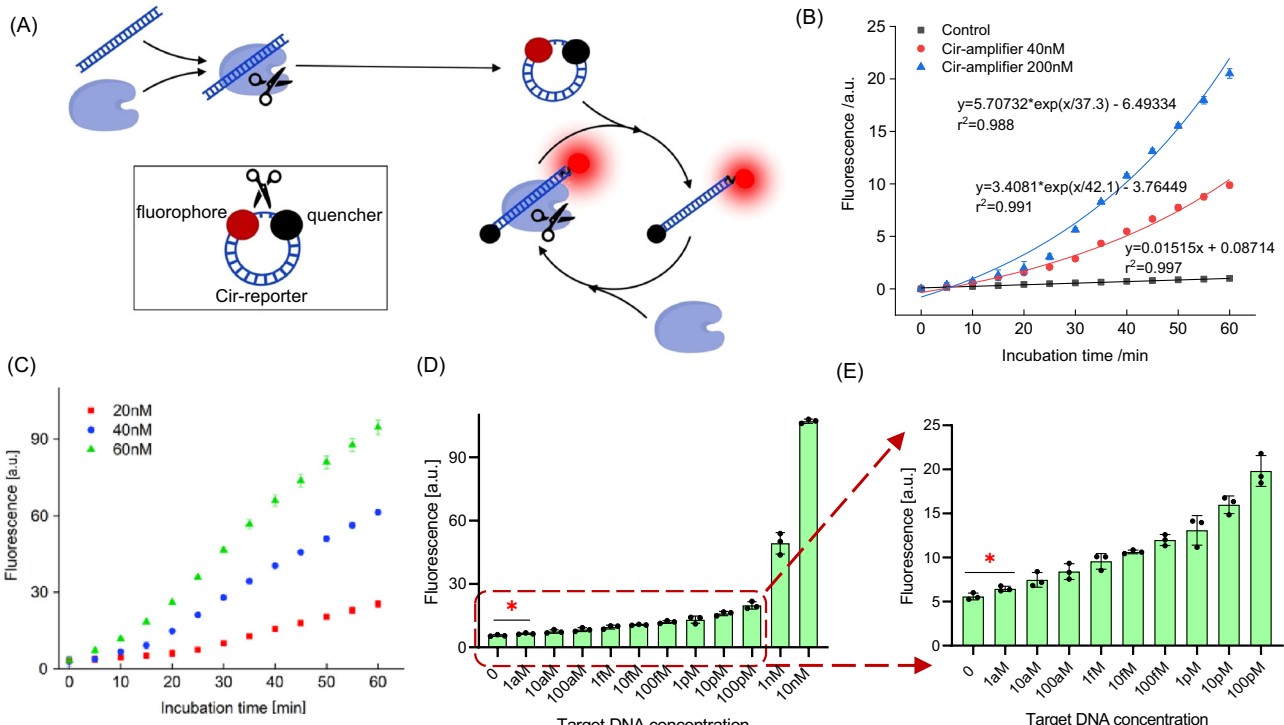

**Fig. 8 | Characterization of the AutoCAR-2 system (Method 17). A** Schematic diagram of AutoCAR-2; **B** The AutoCAR-2 fluorescent signal as a function of Cir-reporter concentration (20 nM of Cas12a RNP, and 1 pM of target DNA) ($n = 3$ independent reactions); **C** The AutoCAR-2 fluorescent signal as a function of Cas12a RNP concentration (200 nM of Cir-reporter, and 1 pM of target DNA) ($n = 3$ independent reactions); **D** The calibration curve of AutoCAR-2 (60 nM of Cas12a RNPs, and 200 nM of Cir-reporter) ($n = 3$ independent reactions). **E** Magnified section of (**D**) at lower concentration range. For the Cas12a concentration used in this work, the assay time is not limited by diffusion time[57] ($n = 3$ independent reactions). (Two tails student T test, Error bars represent mean ± SD, *$P < 0.05$, a.u = arbitrary units).

double-helix (a 20 nt Cir-moiety is ~7 nm long)[44]. High torsional stress is expected due to small radius of curvature along the length of circle (shown by a purple arrow in Supplementary Fig. 35). Furthermore, the closed loop in the Cir-moieties means that they are rotationally constrained, as the initiation of dsDNA unwinding in one location requires increasing of the winding in adjacent locations, and/or in the ssDNA region−unlike in a corresponding linear structure. Thus, a topological barrier in the Cir-moiety prevents it from releasing torsional stress in the perpendicular direction (shown by an orange arrow in Supplementary Fig. 35). At the same time the Cir-moieties are too short to writhe or to supercoil which requires DNA length of ~20 nm or more[45]. Because a Cir-moiety is expected to have a greater torsional stiffness to its corresponding linear form, it requires more energy to unwind and form the required R-loop structure between gRNA and target DNA for Cas12a RNP activation compared to topologically different standard L-dsDNA targets[46]. Thus we propose that bending and torsional stiffness[47] provide intuitions as to possible reasons for reduced binding of Cir-moieties to the RNPs compared with the constituent linear dsDNA.

Based on these unique small circular DNA structures, we developed four autocatalytic sensor schemes which are capable of ultrasensitive detection of a single copy of nucleic acids without amplification at room temperature (Supplementary Table 9). In addition to detecting genomic DNA and RNA at 1 aM levels (~1 copy/μL), the latter without reverse transcription, we were able to quantify oncogenic ctDNA mutations, also at aM levels, including in clinical blood plasma samples using both fluorescent and LFA based colorimetric readout (Fig. 9). Three schemes (AutoCAR 1-3) are single pot reactions, and the most elegant AutoCAR-2 contains only two molecular components, Cas12a RNPs and bifunctional Cir-reporters which activate downstream Cas12a RNPs to drive the autocatalysis reaction system

(Fig. 8) and simultaneously act as fluorescent reporters for signal readout (Fig. 6). We anticipate that AutoCAR may be directly integrated with previously reported type V based CRISPR/Cas biosensing system (based on Cas12 and/or Cas14) as a signal amplifier without any additional changes of the original reagents or setup (Supplementary Fig. 42), as well as with LFAs.

In contrast to conventional nucleic acid amplification technologies such as PCR, RPA, and LAMP[48], AutoCAR provides a comparable sensitivity (1 aM) but without temperature cycling or heating, nor primer issues such as polymerization. Unlike earlier amplification-assisted CRISPR biosensing systems, such as DETECTR[37] (RPA with Cas12a), SHERLOCK[36] (RT-RPA with Cas13), or HOLMES[37] (PCR with Cas12a), AutoCAR offers a rapid amplification-free approach with compatible sensitivity and specificity, but without amplicon contamination, known to be a significant source of false positives. Furthermore, unlike other signal amplification methods used with CRISPR biosensors, such as SERS[49], metal-enhanced fluorescence[50], nanoenzymes, and field-effect transistors[51], AutoCAR does not require any sophisticated components or instrumentation, and it can be performed at room temperature, in a point-of-care setting. In comparison with the recently established Cas tandem[51] and Cas feedback circuit[20] which require more than one hour turnover time due to a complex system design, our AutoCAR system is capable of rapid detection of 1 aM nucleic acids within 15 min (Fig. 8C).

*Trans*-cleavage activity in CRISPR/Cas reactions underpins key applications of CRISPR/Cas systems in bioanalysis, biomedical and diagnostic fields[5,7,12]. Although the pursuit of ortholog or mutant proteins with optimized properties can lead to improvements, the limited *trans*-cleavage represents a challenge for expanding the applications of CRISPR/Cas methods, particularly in biomolecular diagnostics[12,19,52]. Establishing autocatalytic reaction networks provides the key to

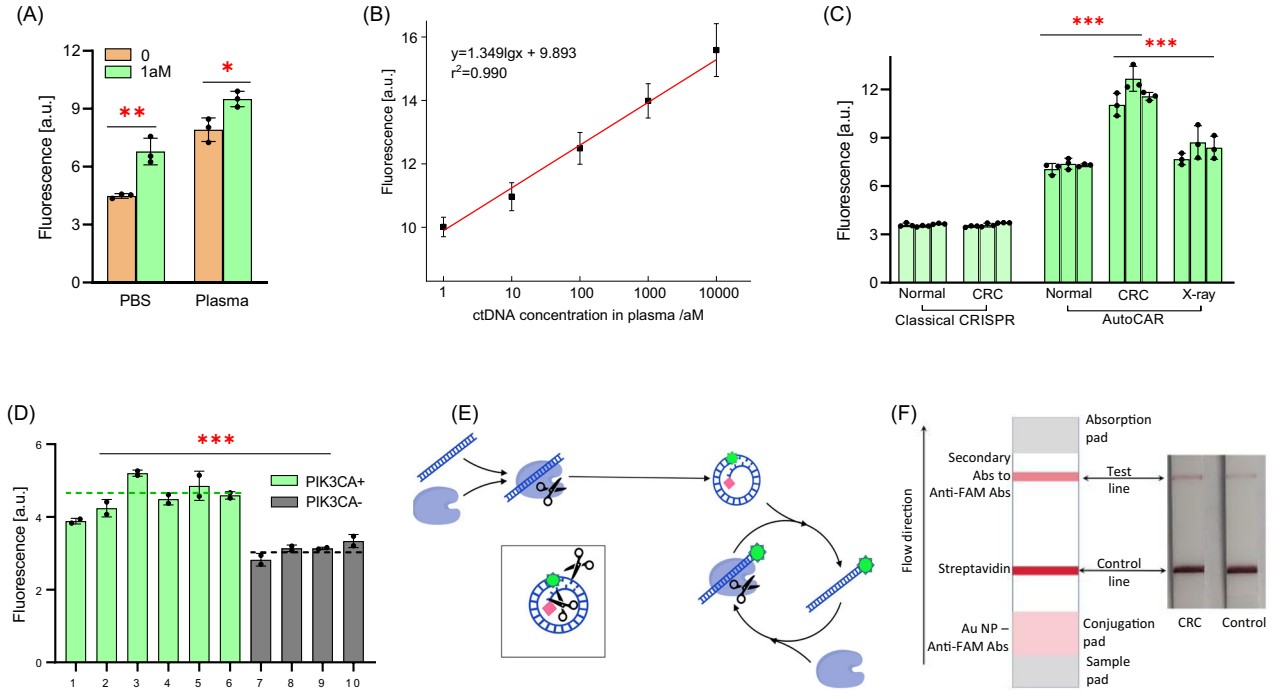

**Fig. 9 | Application of AutoCAR for ctDNA detection from blood plasma (Methods 18, 19, 20, 21). A** Biosensing performance of AutoCAR-3 in PBS and mouse plasma ($n = 3$ independent reactions); **B** AutoCAR-3 calibration curve in mouse plasma ($n = 3$ independent reactions); **C** The application of AutoCAR-3 for ctDNA detection in mouse plasma ($n = 3$ independent reactions); **D** Plasma samples from patients with advanced cancers harboring the PIK3CA H1047R mutation as determined in tumor biopsies (PIK3CA H1047R + $n = 6$, and PIK3CA H1047R - $n = 4$) were subject to detection of circulating PIK3CA mutations in blood plasma using AutoCAR-3 testing. Dashed lines represent the averages of the positive (green) and negative (dark) groups. The feasibility of AutoCAR-3 detection in an alternative body fluid, saliva is shown in Supplementary Fig. 45. **E** Schematic diagram of AutoCAR-4. The establishment of a Cir-color tag includes the extension of cDNA with 5 nt CCCCC and adding biotin on 3' end and adding a FAM fluorophore; **F** The schematic of the colorimetric lateral flow assay (commercial lateral flow strip from Millenia Biotec, with all antibodies pre-integrated) for the detection of Cir-color tag control line: streptavidin; test line: secondary antibody and the application of Cir-color tag-based AutoCAR-4 for ctDNA detection in mouse plasma with a lateral flow assay. The biotin-containing reaction products were captured by streptavidin immobilized on the control line producing color on the control line due to simultaneous presence of Au NPs in these products, as well as FAM and anti-FAM antibodies (anti-FAM Abs). With further flow of the sample, the secondary antibody on the test line captured the anti-FAM antibodies on biotin-free products for colorimetric signal readout. The test line in the CRC LFA (left) shows a darker color compared to the control LFA (right). The background signal in the control LFA is attributed to nonspecific binding or residual Cas12a activation. (Two tails student T test, Error bars represent mean ± SD, *$P < 0.05$, **$P < 0.005$, ***$P < 0.001$, a.u = arbitrary units).

overcome this problem. With its simplicity and minimalistic design, the autocatalytic AutoCAR system reported here promises a pathway to rapid, low cost, and ultrasensitive quantification of nucleic acids, including at point-of-care and in the field. This will expand the boundaries of CRISPR/Cas biotechnology and broaden its applications.

## Methods
### Ethical statement
All animal experiments were approved by the UNSW Animal Care and Ethics Committee (UNSW animal research ethics project approval 20/95B, 21/39B, and 21/77B). NOD/SCID (6–8-week-old) mice were provided by Animal Services from the Animal Resources Centre (ARC, Perth, WA).

All human plasma experiments were approved by the UNSW Ethics Committee (UNSW HC210160), in addition to ACTRN12616000908437. All human saliva experiments were approved by the UNSW Ethics Committee (UNSW HC200568).

### Materials and oligonucleotides
Materials have been listed in Supplementary Information.

All the DNA and RNA oligonucleotides are synthesized and modified by Sangon Ltd. (Supplementary Tables 3–8)

### Synthesis of the Cir-mediator DNA nanostructure (Method 1)
The Cir-mediator was synthesized in a three-step protocol (Supplementary Fig. 4): The Cir-ssDNA was prepared using a reaction mixture containing 2 μL of the linear ssDNA oligo (100 μM), 4 μL of linker ssDNA oligo (100 μM), 2.5 μL of T4 ligase (NEBuffer), 1 μL of T4 ligase buffer, and 43 μL of DI water. The cyclization reaction was allowed to proceed at 16 °C for 12 h and then 65 °C for 10 min, following by storage at 4 °C. Then, the un-ligated linear ssDNA and the linker ssDNA was depredated with a mixture containing 1.5 μL of exonuclease III, 10 μL of Cir-ssDNA product in 40 μL of 1×NEBuffer 2.1 buffer, and incubation at 37 °C for 100 min followed with deactivation of exonuclease III at 75 °C for 30 min. Afterwards, 1:1 molar ratio of cleaned Cir-ssDNA was mixed with its corresponding cDNA at room temperature for 10 min to form the Cir-mediator molecules.

### Standard CRISPR/Cas12a *trans*-cleavage activation assay (Method 2)
1 μL of 100 μM LbaCas12a endonuclease (NEB, M0653T) and 5 μL of 20 μM gRNA was mixed with 3.6 mL 1× NEBuffer 2.1, followed by addition of 6 μL of 100 μM Texas Red quenched reporter. The prepared standard CRISPR/Cas12a reaction mixture was stored at 4 °C for future use. For each CRISPR/Cas12a trans-cleavage activation reaction, 10 μL 10 nM of nucleic acid (L-ssDNA, L-dsDNA, Cir-ssDNA or Cir-

mediator) with complementary sequence to gRNA was added into 90 µL of the prepared reaction buffer. The reaction was carried out at room temperature, and the fluorescence intensity at Ex/Em of 570/615 nm was determined by using a plate reader (iD5 Spectramax, Molecular Devices, USA).

## Urea-PAGE electrophoresis to verify the formation of circular DNA structures (Method 3)

The formation of circular DNA was verified using urea-PAGE electrophoresis. Briefly, 10 µL of DNA circularization product was premixed with 2 µL 6X DNA gel loading dye and then loaded onto premade15% Mini-PROTEANR TBE-Urea Gel (Bio-Rad, 4566055). The electrophoresis was then carried out for 60 min at a constant voltage of 110 V. 2.5 µL of 10 bp DNA ladder was used for molecular weight reference. Gel images were visualized by using Gel Doc + XR image system (Bio-Rad Laboratories Inc., USA).

## Validation of Cas12a trans-cleavage for ssDNA and dsDNA degradation (Method 4)

1 µL of 100 µM LbaCas12a endonuclease (NEB, M0653T) and 5 µL of 20 µM gRNA was mixed with 3.6 mL 1× NEBuffer 2.1, and followed with the adding of 1 µM non-trigger ssDNA or dsDNA oligo. Then, 10 µL of 1 µM trigger ssDNA was mixed with 90 µL of the prepared standard CRISPR/Cas12a reaction mixture. The reaction was set at room temperature for 0–60 mins, and each of 10 µL cleavage product from 0, 10, 20, 30, 40, 50, 60 min was mixed with 2 µL of 6X DNA gel loading dye, and then loaded onto 4% agarose gel for electrophoresis at a constant voltage of 80 V for 60 min. Gel images were visualized by using Gel Doc + XR image system (Bio-Rad Laboratories Inc., USA).

## Re-linearization of Cir-mediators by Cas12a trans-cleavage for further Cas12a activation (Method 5)

1 µL of 100 µM LbaCas12a endonuclease (NEB, M0653T) and 5 µL of 20 µM gRNA was mixed with 3.6 mL 1× NEBuffer 2.1, and followed by the adding of 100 nM of prepared DNA oligo (linear dsDNA, linear ssDNA, Cir-ssDNA or Cir-mediator). Then, 1 µM of trigger ssDNA oligo was added to activate Cas12a trans-cleavage for 60 min at room temperature. Afterwards, 10 µL of the trans-cleavage product was transferred into 90 µL of the standard CRISPR/Cas12a reaction mixture with gRNA matching the Cir-mediator, and set at room temperature for 60 min. The fluorescence intensity at Ex/Em of 570/615 nm was determined by using a plate reader (iD5 Spectramax, Molecular Devices, USA).

## FRET Investigation of interaction of Cir-mediators and Cas12a RNPs (Method 6)

To prepare the FRET assay, the 5′ end of Cas12a gRNA was labeled with a 5′-Cy3 as a donor, and an internal dT at the 3′ of ssDNA (either linear or circularized form) was labeled with a Cy5 as an acceptor (Supplementary Table 3). In brief, the Cy5 labeled ssDNA oligo was firstly mixed with its cDNA to form dsDNA or Cir-mediator, then 5 µL of 1 µM of the prepared DNA oligo was mixed with 95 µL of 1×NEBuffer 2.1 containing Cy3-labeled gRNA loaded Cas12a RNP to a final concentration of 50 nM for both the DNA oligo and Cas12a RNP. The fluorescence signals were then collected by using a plate reader (iD5 Spectramax, Molecular Devices, USA) with Ex/Em of 520/666 nm. The donor-acceptor distance in inactivated and activated Cas12a RNPs were simulated and measured by using PDB 3D Protein Feature View[30]. Briefly, the protein profiles were loaded in the PDB 3D Protein Feature View and then the distance measurement tool was selected. Then, the distance between the corresponding nucleotides labeled with the donor or acceptor fluorophores are measured on the 3D structure model. PDB 5XUS represents activated Cas12a RNP; PDB 6NME represents inactivated Cas12a RNP.

## Preparation of the final AutoCAR reaction mixture (Method 7)

3 µL of 100 µM LbaCas12a endonuclease (NEB, M0653T), 5 µL of 20 µM gRNA1 (for trigger DNA/RNA) and 10 µL of 20 µM gRNA2 (for Cir-mediator) was mixed with 3.6 mL of 1× NEBuffer 2.1 and followed by the addition of 12 µL of 100 µM Texas Red quenched reporter. Afterwards, the prepared Cir-mediator solution was mixed to a final concentration of 50 nM before use to form the final AUTOCAR reaction mixture. Then, 10 µL of different concentrations of trigger DNA (ssDNA or dsDNA) with a complementary sequence to gRNA was mixed with 90 µL of the prepared final AUTOCAR reaction mixture to initiate the reaction. The reaction was set at room temperature and the fluorescence intensity at Ex/Em of 570/615 nm was determined by using a plate reader (iD5 Spectramax, Molecular Devices, USA). Alternatively, Cir-mediator can be replaced by 50 nM of the same volume of 1× NEBuffer 2.1 for a comparison with a standard Cas12a reaction as per Method 2 above.

## Autocatalysis trans-cleavage kinetic in AutoCAR-1 (Method 8)

To assess the kinetics of the AUTOCAR-1 trans-cleavage pattern, the final concentration of 20 nM of LbaCas12a endonuclease (NEB, M0653T), 10 nM of gRNA1 (for trigger ssDNA) and 10 nM of gRNA2 (for Cir-mediator) was mixed with 1× NEBuffer 2.1, along with a constant concentration of 5 nM of target ssDNA and 50 of nM Cir-mediator. In addition, the mixture contained different concentrations of ssDNA reporters (0.156, 0.313, 0.625, 1.25, 2.5, 5 µM). The reactions were conducted at room temperature and the fluorescence intensity at Ex/Em of 570/615 nm was determined by using a plate reader at 120 s intervals (iD5 Spectramax, Molecular Devices, USA).

## Different concentrations of Cir-mediator in AutoCAR-1 (Method 9)

The final concentration of 20 nM of LbaCas12a endonuclease (NEB, M0653T), 10 nM of gRNA1 (matching the trigger ssDNA) and 10 nM of gRNA2 (matching the Cir-mediator) was mixed with 1× NEBuffer 2.1, along with a constant concentration of 1 pM of target ssDNA and 2.5 µM of reporters. In addition, the mixture contained different concentrations of Cir-mediator (0, 2.5, 5, 10, 20 nM). The reactions were conducted at room temperature and the fluorescence intensity at Ex/Em of 570/615 nm was determined by using a plate reader at 120 s intervals (iD5 Spectramax, Molecular Devices, USA).

## AutoCAR-1 for DNA diagnostics on a plate reader (Method 10)

3 µL of 100 µM LbaCas12a endonuclease (NEB, M0653T), 5 µL of 20 µM gRNA1 (matching the target DNA sequence) and 10 µL of 20 µM gRNA2 (matching the Cir-mediators) was mixed at 3.6 mL 1× NEBuffer 2.1 and followed by adding 12 µL of 100 µM Texas Red quenched reporter. Afterwards, the prepared Cir-mediator solution was mixed to a final concentration of 50 nM to form the final AutoCAR-1 reaction mixture for DNA detection. Then, 10 µL of different concentrations of target DNA (ssDNA, dsDNA or *H. pylori* genome DNA) were mixed with 90 µL of the prepared final AutoCAR-1 reaction mixture. The reaction was set at room temperature for 1 h, and the fluorescence intensity at Ex/Em of 570/615 nm was determined by using a plate reader (iD5 Spectramax, Molecular Devices, USA).

## AutoCAR-1 for RNA diagnostics on a plate reader (Method 11)

3 µL of 100 µM LbaCas12a endonuclease (NEB, M0653T), 5 µL of 20 µM gRNA1 (matching the target RNA sequence) and 10 µL of 20 µM gRNA2 (matching the Cir-mediator) was mixed at 3.6 mL 1× NEBuffer 2.1 and followed by adding 120 µL of 100 µM Texas Red quenched reporter and 36 µL of 1 M DTT. Afterwards, the prepared Cir-mediator solution was mixed to a final concentration of 50 nM before the use to form the final AUTOCAR reaction mixture for RNA. Then, 10 µL different concentration of target RNA (RNA or SARS-CoV-2 genome RNA) was mixed with 90 µL of the prepared final c-Car reaction mixture. The reaction

was set at room temperature for 1–1.5 h, and the fluorescence intensity at Ex/Em of 570/615 nm was determined by using a plate reader (iD5 Spectramax, Molecular Devices, USA).

## Gel electrophoresis to verify DNA fragments (Method 12)

The DNA fragments were verified using agarose gel electrophoresis. Briefly, 4% agarose gel in 1×TBE buffer was prepared with SYBR Gold DNA dye (1.5 μL 10,000X into 30 mL agarose gel). 10 μL of DNA fragments were premixed with 2 μL 6X DNA gel loading dye and then loaded onto gel for electrophoresis. The electrophoresis was carried out for 60 min at a constant voltage of 80 V. 1.5 μL of 10 bp DNA ladder was used for molecular weight reference. Gel images were visualized by using Gel Doc + XR image system (Bio-Rad Laboratories Inc., USA).

## Synthesis and characterisation of circular ssDNA on magnetic beads (Method 13)

This method was introduced to improve purity of single ring synthesis and to improve product concentration. The use of magnetic beads makes it possible to concentrate Cir-ssDNA. To synthesize Cir-ssDNA, 0.4 mL of 0.5% w/v streptavidin modified magnetic beads (0.74 μm) were first blocked with 1% BSA solution for 1 h to eliminate non-specific binding. Afterwards, 1 mL of 0.5 μM biotinylated linear-ssDNA was incubated with the beads for 1 h following a PBS wash to remove the residual free linear-ssDNA. Subsequently, 1 mL of the click chemistry reaction solution (1.0 mM CuSO$_4$, 2.0 mM TCEP, and 100 μM TBTA) was added and incubated with the beads for 12 h at room temperature. After synthesis, the magnetic beads were collected and washed with PBS buffer to remove excess chemicals. Subsequently, 100 μL of 100 units/mL Exonuclease VII solution was added and incubated at 37 °C for 30 min to remove the linear ssDNA. After washing with PBS buffer, the synthesized Cir-ssDNA was released from the streptavidin-modified magnetic beads by heat treatment at 95 °C for 30 min[53], and the supernatant was collected for further use. All the Cir-ssDNA used in this research are synthesized based on this approach. The sequence listed in Supplementary Table 4 is a demonstration example. Nanodrop was utilized to test the concentration of synthesized Cir-ssDNA.

The formation of Cir-ssDNA was verified by using denaturing polyacrylamide gel (dPAGE) electrophoresis assay. 10 μL of Cir-ssDNA aliquoted with 2 μL 6X DNA gel loading dye was loaded into the gel for electrophoresis, which was carried out for 40 min at a constant voltage of 100 V. 5 μL of 10 bp DNA ladder was used for molecular weight reference. Gel images were visualized by using Gel Doc + XR image system (Bio-Rad Laboratories Inc., USA).

## Investigation of the reporter performance of Cir-reporters in a standard CRISPR/Cas12a biosensing system (Method 14)

The Cir-reporter was assembled by mixing Cir-ssDNA with fluorophore labeled cDNA (Texas-cDNA-BHQ2) at a ratio of 1:1. The Cir-reporter based CRISPR/Cas12a reaction mixture was prepared as follows: 1 μL of 100 μM (100 pmol) of Cas12a protein was gently mixed with 5 μL of 20 μM (100 pmol) of gRNA-C in 3.6 mL 1X NEB 2.1 buffer. Then, 120 μL of 5 μM (0.6 nmol) of Cir-reporter solution with different linker length (0–7 nt) was added and well mixed to form the standard Cir-reporter-assisted reaction mixture. For comparison, linear ssDNA reporter-assisted CRISPR/Cas12a reaction mixture was prepared as follows: 1 μL of 100 μM (100 pmol) of Cas12a protein was gently mixed with 5 μL of 20 μM (100 pmol) of gRNA-C solution in 3.6 mL 1X NEB 2.1 buffer. Then, 6 μL of 100 μM (0.6 nmol) of pre-synthesized fluorescent quenched ssDNA reporters (Texas red-TTATT-BHQ2) was added and well mixed to form the standard linear ssDNA reaction mixture.

Afterwards, 10 μL of different concentrations (0, 0.1, 1, 10, 100, 1000 pM) of target-C ssDNA were added to 90 μL of the prepared reaction mixture containing either Cir-reporters or ssDNA reporters and incubated for 120 min. A SpectraMax iD5 multi-Mode Microplate

Reader (Molecular Devices) was used for the detection of fluorescence readout. The Ex/Em of Texas-Cir-reporter-BHQ2 was 570/615 nm. All DNA and RNA oligos used in this experiment are listed in Supplementary Table 4.

## Investigation of the RNP activation ability of Cir-reporter in a standard CRISPR/Cas12a biosensing system (Method 15)

In this experiment, the CRISPR/Cas12a reaction mixture was prepared as follows: 1 μL 100 μM (100 pmol) of Cas12a protein was gently mixed with 5 μL 20 μM (100 pmol) of gRNA-D in 3.6 mL 1X NEB 2.1 buffer. Then, 6 μL of 100 μM (0.6 nmol) of pre-synthesized fluorescent quenched ssDNA reporters (Texas red-TTATT-BHQ2) was added and well mixed to form the standard reaction mixture.

Afterwards, 10 μL of 0.25 μM of Cir-reporters with different dsDNA length and different ssDNA linker lengths were added to 90 μL of the prepared reaction mixture and incubated for 120 min. A SpectraMax iD5 multi-Mode Microplate Reader (Molecular Devices) was applied for the detection of fluorescence readout. The Ex/Em of Texas red-TTATT-BHQ2 reporter was 570/615 nm. For comparison, linear dsDNA was also applied to activate the CRISPR/Cas12a reaction mixture under the same conditions. All DNA and RNA oligos used in this experiment are listed in Supplementary Table 5, 6.

## Investigation of the RNP activation efficiency of linearized Cir-reporters in a standard CRISPR/Cas12a biosensing system (Method 16)

The RNP activation efficiency of linearized Cir-reporters was evaluated using the CRISPR/Cas12a reaction mixture prepared by Method 15. Afterwards, 10 μL 0.25 μM of linearized Cir-reporters was added to 90 μL of the prepared reaction mixture and incubated for 120 min at room temperature. A SpectraMax iD5 multi-Mode Microplate Reader (Molecular Devices) was applied for the detection of fluorescence readout. For comparison, linear dsDNA (18nt) was also applied to activate the CRISPR/Cas12a reaction mixture under the same conditions. All DNA and RNA oligos used in this experiment are listed in Supplementary Table 5.

## Evaluation and biosensing application of AutoCAR-2 (Method 17)

The AutoCAR-2 reaction mixture was prepared as follows: 1 μL of 100 μM (100 pmol) of Cas12a protein was gently mixed with 5 μL of 20 μM (100 pmol) of gRNA-D to form the Cas12a RNP in 5 mL 1X NEB 2.1 buffer. Subsequently, 200 μL of 5 μM (1 nmol) of Cir-reporter solution was added and well mixed to form the reaction mixture.

Afterwards, 10 μL of target-D ssDNA at different concentrations was added to 90 μL of the prepared reaction mixture for activation of *trans*-cleavage of Cas12a and enabling the CRISPR/Cas autocatalysis biosensing reaction. A SpectraMax iD5 multi-Mode Microplate Reader (Molecular Devices) was applied for the detection of fluorescence readout. The Ex/Em of Texas red-TTATT-BHQ2 reporter was 570/615 nm. All the DNA and RNA oligos used in this experiment are listed in Supplementary Table 7.

## Establishment of orthotropic CRC mouse model (Method 18)

All animal experiments were approved by the UNSW Animal Care and Ethics Committee (UNSW animal research ethics project approval 20/95B, 21/39B, and 21/77B). NOD/SCID (6–8-week-old) mice were provided by Animal Services from the Animal Resources Centre (ARC, Perth, WA). Mice were housed in specific pathogen free conditions at 22 °C with a light/dark cycle of 12 h. Mice were kept in standard ventilated cages and acclimated for one-week following arrival into the UNSW animal facility. Mice were provided food and water and libitum, and their wellbeing was monitored regularly.

An orthotropic CRC mouse model was established by using intra-rectal tumor cell injection method with minor modifications of the

previously reported work[54]. In brief, 6–8-week-old female NOD/SCID mice were fasted of food for 6 h prior to cancer cell injections, followed by rapid anaesthesia induction with 2–4% isoflurane and maintenance at 1–3% with 1 L/min oxygen. Lubricated blunt-tip forceps were used to dilate the anal canal, exposing the distal anal and rectal mucosa. Subsequently, $4 \times 10^5$ HCT-116-Luc2 cells (CCL-247-LUC2, ATCC) suspended in 10 μL PBS and 10 μL Matrigel were orthotopically inoculated into the distal posterior rectal submucosa, 1–2 mm above the anal canal using a 30-gauge needle (Terumo, Tokyo, Japan). Mice were closely monitored for 1 to 72 h post-injection for early detection of adverse events, with subsequent monitoring occurring at least bi-weekly.

Tumor formation and growth over time were monitored once a week by using the IVIS Spectrum CT imaging system (Perkin Elmer, Waltham, US). Typically, mice were intraperitoneally injected with 150 mg/kg of D-Luciferin. Mice were then anesthetized with isoflurane, with anaesthesia maintained throughout imaging using the IVIS spectrum imaging system for bioluminescence detection via Living Image® 4.5.2 software. When tumor reached the 100 mm³ volume (equivalent to approximately $4-6 \times 10^{10}$ photons/s of bioluminescence signal in this study), one group of mice were treated with X-ray radiation. At 27 days post treatment, the terminal blood collection (500–750 μL per mouse) was performed by the cardiac puncture technique with 25-gauge needles. K3 EDTA tubes were used for blood samples collection, allowing the isolation of blood plasma through centrifugation ($1000 \times g$, 10 min). The isolated mice blood plasma was stored at −80 °C for further use.

### Application of AutoCAR-3 for the detection of ctDNA in mouse plasma, human plasma and human saliva (Method 19)
The AutoCAR-3 reaction mixture for ctDNA (PIK3CA H1047R) detection was prepared as follows: 1 μL of 100 μM (100 pmol) of Cas12a protein was gently mixed with 5 μL of 20 μM (100 pmol) of gRNA-ct to form the Cas12a RNP-1. In the meanwhile, 1 μL of 100 μM (100 pmol) of Cas12a protein was gently mixed with 5 μL of 20 μM (100 pmol) of gRNA-D to form the Cas12a RNP-2. Afterwards, the prepared Cas12a RNP-1 and Cas12a RNP-2 were mixed with 200 μL of 5 μM (1 nmol) of Cir-reporters in 5 mL 1X NEB 2.1 buffer to form the standard reaction mixture.

Afterwards, 10 μL of spiked in sample or collected mouse plasma, human plasma or human saliva was added to 90 μL of the prepared reaction mixture for activating trans-cleavage of Cas12a and enabling the CRISPR/Cas biosensing reaction. A SpectraMax iD5 multi-Mode Microplate Reader (Molecular Devices) was applied for the detection of fluorescence readout. The Ex/Em of Tex-Cir-reporter-BHQ2 was 570/615 nm. All the DNA and RNA oligos used in this experiment are listed in Supplementary Table 8.

All human plasma experiments were approved by the UNSW Ethics Committee (UNSW HC210160), in addition to ACTRN12616000908437. All human saliva experiments were approved by the UNSW Ethics Committee (UNSW HC200568).

### Application of AutoCAR-3 for ctDNA detection in clinical plasma samples (Method 20)
Patient bloods from The Cancer Molecular Screening and Therapeutics (MoST) program were assessed (ACTRN12616000908437). The study was performed in accordance with the Declaration of Helsinki. The program has been approved by the St Vincent's Hospital Sydney Human Research Ethics Committee (reference, HREC/16/SVH/23). All patients provided written informed consent for participation in this study. Eligibility criteria included patients with advanced solid cancers of any histological type, prioritizing rare cancers.

For each patient enrolled in the MoST program, 10 sections of 4 μm of formalin fixed paraffin embedded (FFPE) tissue underwent DNA extraction. Tissue deparaffination was performed using a

deparaffinization solution and DNA extraction using AllPrep DNA/RNA kit (Qiagen, Germantown, MD, USA). Quantification was performed using Qubit double-stranded DNA (dsDNA) high-sensitivity (HS) assay kit (Invitrogen, Waltham, MA, USA).

CGP profiling was undertaken using FFPE archival tumor[55]. Patients were screened at entry into the MoST program. The panel-based assays employed included Illumina TruSight Tumor 170, Illumina TruSight Tumor 500, or the Foundation Medicine (FMI), and conducted using standard protocols. All three panels had PIK3CA H1047R mutation detection capabilities.

Venous blood -10 mL was collected by standard phlebotomy techniques into anticoagulant containing tubes. The time from blood collection to processing was 0–5 days (median: 2 days). Blood was kept at room temperature before and during processing. Bloods were centrifuged at $1000 \times g$ for 10 min at 4 °C, after which plasma was transferred to a new tube and spun for an additional 10 min at $1000 \times g$, 4 °C. Plasma was removed leaving behind any residual pellet and aliquots stored at −80 °C until assayed.

For this study, plasma samples of ten patients ($n = 6$ positive and 4 negative for PIK3CA H1047R mutation) were tested by using Method 19.

### Application of colorimetric Cir-color tag based AutoCAR-4 for ctDNA detection in mouse plasma using lateral flow assay (Method 21)
The Cir-color tag based AutoCAR-4 reaction mixture was prepared as follows: 1 μL of 100 μM (100 pmol) of Cas12a protein was gently mixed with 5 μL of 20 μM (100 pmol) of gRNA-ct to form the Cas12a RNP in 5 mL of 1X NEB 2.1 buffer. Subsequently, 200 μL of 5 μM (1 nmol) of Cir-color tags were added and well mixed to form the reaction mixture.

Afterwards, 10 μL of the collected mouse plasma was added to 90 μL of the prepared reaction mixture for activating trans-cleavage of Cas12a and enabling the CRISPR/Cas biosensing reaction. After 15 min reaction, 5 μL of the reaction mixture was added to 95 μL of Hybri-Detect assay buffer (Millenia) and run on HybriDetect lateral flow strips (Millenia). All the DNA and RNA oligos used in this experiment are listed in Supplementary Table 8.

### Additional Methods
All additional Materials and Methods including Statistical Methods have been listed in Supplementary Information.

### Reporting summary
Further information on research design is available in the Nature Portfolio Reporting Summary linked to this article.

## Data availability
Source data are provided with this paper. No restriction on data availability. PDB data used in this study including: 5XUS, 6NME. Source data are provided with this paper.

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

## Acknowledgements

The authors acknowledge the support of the ARC Centre of Excellence for Nanoscale Biophotonics (CE14010003), UNSW SHARP program of E.M.G., funding (GNT1181889) from the Australian National Health and Medical Research Council, fellowship award of W.D. (2019/CDF1013) from Cancer Institute NSW, Australia and project grant from National Foundation for Medical Research and Innovation, Australia. D.M.T. is supported by a National Health and Medical Research Council L3 Investigator grant (GNT1195742). We acknowledge Dr Carl Power from Biological Resources Imaging Laboratory, UNSW for performing the orthotopic injection of CRC cells into mice. We acknowledge Dr Sheri Nixdorf from University of Technology Sydney and Dr Tzongtyng Hung from UNSW for providing guidance on monitoring the tumor development and determining the tumor size. Patient cancer blood samples were collected under an approved research study (ACTRN12616000908437) with funding provided by Medical Research Futures Fund, and the New South Wales Office for Health and Medical Research.

## Author contributions

F.D contributed ideas design, data collection and analysis, and manuscript preparation. Y.L contributed ideas design, data collection and analysis, and manuscript preparation. B.Y contributed data collection and analysis. R.S and W.D. contributed the animal model. M.K, D.M.T, S.T, and F.L. contributed to the clinical data. E.G contributed ideas design and the theoretical model. All authors contributed to and approved the final manuscript.

## Competing interests

The authors declare no competing interests.
