## [Peer Review File · Nature Communications]

Reviewers' Comments:

Reviewer #1:

Remarks to the Author:

In this manuscript, the authors describe an autocatalytic CRISPR/Cas12a amplification Reaction (AutoCAR) for sensitive nucleic acid detection. The AutoCAR system utilizes Cas12a ribonucleoprotein (RNP) to trans-cleave the designed cir-mediators, which activate multiple RNPs in turn, resulting in signal amplification for target nucleic acid detection. The authors apply the AutoCAR system to detect genomic DNA (*H. pylori*) and RNA (SARS-CoV-2) diagnostics at 1 aM level sensitivity. However, upon careful examination of the data, several issues were found that may question the fundamental design of the work. Therefore, these issues need to be fully addressed, and major revision is recommended before consideration by Nature Communications.

1. The statement in the abstract, "increase the achievable trans-cleavage rates per target by over 3 orders of magnitude," is misleading because this paper used the trans-cleavage of Cas12a RNP to activate more RNPs that generate amplified signals, and not to significantly alter the trans-cleavage rate of Cas12a.
2. Previous literature has revealed that Cas12a trans-cleaves ssDNA into small fragments with lengths of 2 or 4 nt (Cell Res, 2018, 28, 491). Since the cir-mediator used in this paper contains only a 2 nt single-stranded region, direct evidence is needed to demonstrate whether it can be trans-cleaved by an activated Cas12a RNPs.
3. The cir-mediator used in this paper (Fig 1B) is cleaved to form an 18 bp dsDNA, but only a 14 nt sequence can hybridize with gRNA spacer. However, the results in Fig S3 showed that the 18bp dsDNA induced only a very weak signal, slightly above the negative control. Therefore, it is difficult to imagine how an effective chain activation could be generated using the cir-mediator.
4. The concentrations of trans-cleaved reporters in Fig. 4 D-F are significantly higher than those of the reporters used in the experiments. As an example, in Fig. 4E, the maximum concentration of ssDNA reporter is 5 μM , while the concentration of cleaved reporters can reach 20 μM according to the y-axis. Why is there such a huge difference?
5. The blank signal in Fig. 5 exhibits a significant variation between 0.8 and 4, which requires a comprehensive discussion. Moreover, the marker curve indicates an increase in signal with increasing concentration, but the actual samples saturate at approximately 10 copies/ μL , and the signal decreases even at higher concentrations. This phenomenon needs to be explained.
6. It is important to clearly indicate whether synthetic genome fragments or real pathogenic genomic DNA/RNA were used during ultra-sensitive DNA and RNA detection. Furthermore, the practical application of the method should be demonstrated using real samples in excess of 50.
7. In the AutoCAR system, the amount of activated Cas12a RNPs increases continuously. It is challenging to accurately determine the Michaelis-Menten constant in this system due to its complex dynamics.

Reviewer #2:

Remarks to the Author:

The manuscript by E. M. Goldys and colleagues, entitled "Topological barrier to Cas12a activation by circular DNA nanostructures facilitates autocatalysis," reports on the development of an autocatalytic CRISPR-Cas12a-based detection system (AutoCar) that takes advantage of the topological barrier to Cas12 activation due to circular DNA nanostructures. The authors demonstrate that the activation of Cas12a trans-cleavage can be selectively restricted by a special design of a small circular DNA nanostructure that partially matches the gRNA sequence. The proposed detection system takes advantage of two distinct RNP complexes, each characterized by a unique RNA guide. The first RNP (RNP1) is associated with target recognition, while the second RNP (RNP2) is directly involved in the signal amplification process because it is complementary to the sequence of an engineered circular dsDNA molecule (i.e., Cir-Mediator). The amplified output of trans-cleavage products is attributed to the increased number of Cas12a RNP complexes (RNP2) that can be activated by linearization of circular dsDNA nanostructures in response to target activation of RNP1. The near-exponential trend of substrate cleavage products confirms the presence of autocatalysis with an increase in achievable trans-cleavage rates per target of over three orders of magnitude. Compared to the literature, the authors report that the AutoCar system enables ultrasensitive CRISPR-based detection of nucleic acid targets (DNA, RNA).

The manuscript is interesting and well-organized, and the sensing approach is intelligent. The use of topology constraints to control trans-cleavage activity is novel for the field of CRISPR-Cas-based diagnostics of nucleic acids. However, despite the good degree of novelty demonstrated in the system, the manuscript presents significant methodological issues that need to be clearly addressed to make it suitable for publication in Nature Communications.

1) The model and experimental data support the fact that the approximate number of activated Cas molecules per single target molecule (R cas/target) has increased by three orders of magnitude in the AutoCar system (Table S2, reaction time of 1500 s). However, this seems inconsistent with the reported aM detection limit reported in figure 5 for DNA and RNA detection. Indeed, aM detection limits mean approximately six orders of magnitude lower than the well-established LoD reported for pre-amplification free Cas12a detection systems (1-10 pM). How is it possible to achieve six orders of magnitude lower LoD with a three orders of magnitude increase in activated Cas12 enzyme per target? The authors should critically analyze and deeply discuss this point in the main text since it cannot be simply ascribed to the longer time of analysis (1-1.5 h, see methods 4.12 and 4.13) that is longer compared to the time of the assay reported for the generation of the model (1500 s, 25 minutes).

2) In addition, the authors should add two experimental panels (one for DNA and one for RNA detection) in figure 5 reporting the raw kinetic profiles (fluorescence vs. time) of the trans-cleavage activity - with appropriate control experiments - in the range of concentrations (from aM to fM) to clearly demonstrate the reported sensitivity.

3) Ensemble sensors rely on a "large" sensing volume, leading to a response change that is the aggregate response from the sensor. To achieve ultra-low concentration sensitivity, as has been clearly reported recently (see ACS Sens. 2023, 8, 3, 941-942), sampling issues are important for concentrations below fM, where the typical sample volumes of μ L do not ensure an ample supply of analyte. Hence, the authors should highlight if the experimental procedures here reported meets this requirement.

4) The study lacks specificity tests, which is a significant limitation. The authors should characterize the AutoCar system with single and double MM DNA targets to determine if the platform is specific to the target sequence.

5) The study appears to be incomplete in terms of statistical analysis. Although many histograms are presented, the associated data points and number of replicates should be reported. Image captions should include the number of replicates performed in each experiment, and when P values are defined, the authors should specify how they were calculated.

Minor points to be addressed:

1) In most of the assays, a concentration of 50 nM Cir-mediator is used, but in Figure 4 panel F, this concentration is not investigated. The authors should add this data to the panel, and the raw fluorescence data associated with panel F should be included in the supporting information.

2) Figure 2B shows fluorescence vs ssDNA length, not the trans-cleavage rate, so the text should be revised accordingly.

3) The colors of the lines and the legend in Figure S6 are not very clear.

4) In Figure 2G, the authors should explain the drift observed in the gel electrophoresis of dsDNA sequences. They should also demonstrate that Cir-mediator is not cleaved over time using gel electrophoresis, not only the "linear" dsDNA sequence as reported.

5) Section 7 reports that the experimental method used to analyze the *H. pylori* glm gene fragment is 4.11, but the image caption indicates that it is method 4.12. The authors should check all the text and image captions to ensure consistency regarding the methods mentioned.

6) The caption of Figure S6 does not include information about the length of the ssDNA sequence, its concentration, or other experimental conditions. The authors should improve the clarity of their experimental conditions.

7) For the data in Figure S17, it is not clear whether a target activator of the ssDNA or dsDNA was used or at what concentrations. The authors should clarify these experimental details.

Reviewer #3:

Remarks to the Author:

The authors, Li et al., of the article "Topological barrier to Cas12a activation by circular DNA nanostructures facilitates autocatalysis", have developed a circular DNA target ("Cir-mediator") for CRISPR-Cas12a that can successfully enhance the nucleic acid detection sensitivity to attomolar ranges leveraging trans-cleavage in Cas12a. This Cir-mediator target comprises of a single-stranded DNA region for trans-cleavage cut, and a double-stranded DNA region consisting of the PAM sequence and crRNA complementary region. The basis of this ingeniously designed Cir-mediator trans-cleavage DNA targets is that its addition in the detection solution does not self-initiate the activation of CRISPR-Cas12a protein, owing to bending and torsional stiffness. Thus, cleavage of the Cir-mediator target via trans-cleavage upon the activation of the protein by the target nucleic acid prompts it to an autocatalytic feedback loop reaction mechanism (AutoCAR). This enhances the overall detection of the target by deploying activated Cas12a for trans-cleavage of the reporter DNA.

The authors have appreciably investigated the length and breadth of the subject to discard a multitude of different possibilities for non-specific detection. They also categorically showed how the AutoCAR mechanism follows a near-exponential trend and categorically addressed that the relation with the target nucleic acid to be detected is linear. They demonstrated that the proposed mechanism also successfully detects ssRNA targets at attomolar ranges without requiring reverse transcription or any amplification strategies.

Overall, the proposed application is excellent and greatly valuable for nucleic acid detection. Minor issues in the paper are identified as follows:

1. The authors should show the cleaved product band in the ssDNA gel and its absence in the dsDNA gel, or else quantify them (Fig. 2G).

The authors, Deng et al. and Li et al. presented two research articles: "Topological barrier to Cas12a activation by circular DNA nanostructures facilitates autocatalysis" and "Bifunctional circular DNA amplifier transforms a classic CRISPR/Cas sensor into an ultrasensitive autocatalytic sensor". Both articles are from the same lab of Dr. Ewa M. Goldys, concerning the use of an autocatalytic mechanism in Cas12a which has been ingeniously transformed into an extraordinarily rapid diagnostic tool for both DNA and RNA detection. While the former article deals with the basic science behind the designing of the AUTOCar scheme, the latter details the design of the biosensor. In order to be fair, the mechanism illustrated in both the articles are the same, but the purpose and intention have been laid out differently. While the former delved into great details of the claim and demonstrated the functioning and success of the technique by providing extensive experimental reports, the latter is more focused on the designing of the biosensor and showcased its sensitivity and performance.

Both articles are excellent, and the results deserve to be published in a high-impact journal. However, considering the overlap in the experimental strategies and the results shown, it would be ideal to condense the two papers into one research article.

Response to reviewers' comments

In response to suggestions from editors and reviewers, we have now combined the two manuscripts into one. Correspondingly, all figure and table numbers used in the responses to reviewers now refer to the newly integrated manuscript.

Responses to the comments for the manuscript "Bifunctional circular DNA amplifier transforms a classic CRISPR/Cas sensor into an ultrasensitive autocatalytic sensor" (manuscript NCOMMS-23-11654-T, now closed)

The manuscript corrections related to NCOMMS-23-11654-T have all been transcribed to the integrated manuscript, and marked by **yellow** highlights.

Reviewer #2

Reviewer #2 Comment 1. In particular, the authors claim their assay is ultra-rapid (15 min) and highly specific. Regarding the response time, it is well-established that response time is a huge challenge for ultrasensitive sensors because of the limitation of mass transport (i.e. getting the analyte to the sensor). Generally speaking, binding kinetics or detection events become slower at lower concentrations, especially in the absence of mixing or enhanced mass transport. For instance, in the presence of 1fM of analyte concentration, the time between effective binding of ctDNA to Cas12 enzyme would be of the order of approx. 10 min in diffusive regime (see *J. Am. Chem. Soc.* 2019, 141, 3, 1162–1170). Thus, the authors should provide a coherent explanation for their capacity to achieve aM detection limits in homogeneous solution within 15 minutes of analysis.

Our response: This comment is likely motivated by the perspective of interfacial sensors (see *J. Am. Chem. Soc.* 2019, 141, 3, 1162–1170) where sensing occurs on a surface – the case that is materially different to the case of a solution-based sensor such as ours. In solution-based CRISPR/Cas sensors, the target molecule needs only to diffuse to the nearest Cas12 molecule – which is a much shorter travel distance than to a surface. That distance depends on our concentration of Cas proteins, which is under our control. For example, for Cas12a concentration of 20 nM used in some of our plots (one of the lower values we used), the average distance between the Cas12a proteins is ~ 0.1 mm, and diffusion of a nucleic acid target to the nearest Cas12 protein will require the time of ~10 s to 70 s for the diffusion constants in the range $5 \times 10^6 \text{ cm}^2/\text{s}$ (if target is the more rapidly diffusing molecule, having molecular weight in the order of 300 Daltons) to $6.9 \times 10^{-7} \text{ cm}^2/\text{s}$ (if Cas12a is the more rapidly diffusing molecule, having molecular weight in the order of 100k Daltons). These times are much lower than 15 mins, and can be further reduced by increasing the concentration of Cas12a.

In order to address the reviewers request to provide a coherent explanation of 15 min 1 aM assay in light of the limitation of mass transport, and to clarify potential misconceptions, we added a comment in the caption to Figure 8E which reads “For the Cas12a concentration used in this work, the assay time is not limited by diffusion time”. We also added the relevant reference “Wu, Y., Tilley, R. D., & Gooding, J. J. Challenges and solutions in developing ultrasensitive biosensors. *Journal of the American Chemical Society*, 141(3), 1162-1170 (2019)” to the revised manuscript.

Reviewer #2 Comment 2. Other critical issues include the potential sampling issue for ultra-low concentration sensitivity,

Our response: We acknowledge that the distribution of molecules in small samples at low number of molecules may lead to sampling inaccuracies. The distribution of molecules in such small samples is Poissonian. However, the effect of statistically-driven sampling inaccuracies becomes less and less pronounced as the number of molecules in the sample increases. This is illustrated in the figures below which we prepared for this reviewer (not included in the manuscript), where the horizontal axis represents the number of molecules in the sample and the vertical axis is the probability of finding this number of molecules in that sample according to the Poisson distribution. As an example, the middle panel shows that for a 10 µL sample size, 80% of samples have from 15 to 25 molecules

(notwithstanding any pipetting inaccuracies). In order to moderate this effect, all measurements were made in triplicate which was mentioned before, but is now explicitly stated in our added Statistical Methods.

In order to address this comment (and a similar comment from another reviewer) we added a new Statistical Methods section in the revised Supplementary Information (Method S19). The added text reads “Our experimental procedure of measuring molecular concentrations and related quantities in triplicate is not significantly affected by sampling issues at molecular concentrations and sample volumes used in this study, despite their low values. All our measurements are taken in triplicates (X_1, X_2, X_3) and then we use the average value of readings X . If X represents the experimentally obtained number of molecules (the “sample estimate” in the language of statistics), it is possible to calculate the probability that this X represents the real average number of molecules, m , in these samples with 20% accuracy. To this aim we need calculate the probability $P(0.8X < m < 1.2X)$. This probability is the same as the probability $P(10/12 m < X < 10/8 m) = P(10/4 < X_1 + X_2 + X_3 < 30/8 m)$.

The random variable $X_1 + X_2 + X_3$ has a Poissonian distribution with the constant $\lambda = 3m$.

Therefore $P = \sum_{\frac{10m}{4}}^{\frac{30m}{8}} \frac{(3m)^k}{k!} e^{-3m}$. Using this expression, we calculated the applicable numerical values of probability P .

For 1 aM samples at 10 µL volume used in this work for plasma measurements we have $m=10, \lambda=30, 30m/8=37.5, 10m/4=25$, producing $P=0.75$. For 10 aM samples at 10 µL volume, $m=100, \lambda=300, 30m/8=375, 10m/4=250$, leading to $P=0.998$. For larger volumes and higher concentrations P is higher than 0.998. All these probabilities are close to unity which means that our sample estimate X (the average of X_1, X_2, X_3) closely represents the value of m .”

The above calculation can be easily modified to describe measurement accuracy different than 20%.

Reviewer #2 Comment 3 the lack of any specificity tests for targets with single base mismatch,

Our response: We emphasise that our autocatalytic CRISPR sensor inherits its specificity from standard CRISPR sensors which is well established in the literature. In order to address this comment,

we added new data on synthetic targets with single base mismatch (Supplementary Information Figure S40) referred to by an added phrase in the manuscript in Section 10: “The detection of single nucleotide mismatches in the target sequence has also been demonstrated (Figure S40).” In addition, the clinical data we added to Figure 9D and S44 also demonstrate that AutoCAR has single base specificity.

Reviewer #2 Comment 4: incomplete statistical reporting (experimental points and the number of replicates are not always reported in the histograms).

Our response: The manuscript does not contain any histograms, so we assume here that the reviewer meant “bar charts”. Bars in our bar charts represent fluorescence readings for triplicate samples. The height of the bar represents the average value of these 3 readings and the error bar on each represents the standard deviation of these readings, as standard in scientific literature. This information has been added to the new Statistical Methods section in Method S19.

Reviewer #3:

Moderate revisions for publication in Nature Communications include:

Reviewer #3 Comment 1. Statistically relevant positive and negative predictive agreements should be shown.

Our response: We have now added this information to the figures throughout the whole manuscript, each time the statements about positive and negative agreements were made. The added Statistical Methods section describes the statistical testing used and explains the relevant symbols (* $P < 0.05$, ** $P < 0.005$, *** $P < 0.001$, ns=non-significant) utilised in all necessary Figures.

Reviewer #3 Comment 2. As per Fig. 5c, there is no statistical significance of the increment in fluorescence between 1 aM and 10 aM, and neither between 10 aM and 10 aM. In a real experimental situation, will the method be able to distinguish between the target concentration, especially because it holds clinical significance.

Our response. We remind that figure numbering in the integrated manuscript is now changed due to the combination of two manuscripts, and the figure referred to by the reviewer is now Figure 8D and 8E.

We appreciate the clinical significance of precise readings at the lower end of assay range. Our data show that it is possible to adjust this detection range by varying the concentration of reagents and the assay scheme used (noting that in this revised integrated manuscript we proposed 4 different schemes for the AutoCAR assays (AutoCAR 1-4) to choose from, varying detection range, including the ranges listed by the reviewer...

The conditions allowing ultrasensitive detection in the range mentioned by this Reviewer are shown in Figure 5A. This figure illustrates the detection of targets over 3 orders of magnitude range allowing

to distinguish between 1 aM and 10 aM, between 10 aM and 100 aM and between 100 aM and 1000 aM. The conditions of the assay in Figure 8D and 8E were chosen to facilitate a broad detection range of over 11 orders of magnitude. This has advantages in alternative applications such as microbiology or infectious disease diagnostics, where target concentrations may span a wide range – but more limited distinction between close values of target concentration may not be very important.

Reviewer #4

Reviewer #4 Comment 1. In Figure 3, the authors compared the signal output performance of a typical linear 5nt-ssDNA reporter and a Cir-amplifier that was optimized with a 3-nt ssDNA linker. However, the distances between the fluorophores and quenchers of these two systems are different (5nt vs 3nt). It is better to provide a comparison with same length.

Our response: In Figure 3E (currently renumbered as Figure 6E in combined manuscript), we compared the fluorescence signal for a linear 5nt-ssDNA reporter and Cir-reporters with a 3nt ssDNA linker. In Figure 6D, we compared the fluorescence signal of Cir-reporters with 3nt ssDNA linker and Cir-reporters with 5nt ssDNA linker, and no significant differences between them were observed. Therefore, the results in Figure 6E (5nt linear vs 3nt Cir-reporters) are equivalent to the comparison of these two reporters for the same linker length.

Reviewer #4 Comment 2. Compared with classic system, DANCER achieved 1 aM level DNA detection, which is 10⁶ times improvement. The mechanism of such significant improvement should be discussed more deeply.

Our response: In this new integrated manuscript, the establishment and discussion of the mechanism is the main focus of the vast majority of its content, in the first half of the manuscript, and also in the Supplementary Note that provides the theoretical model. We trust that this combined manuscript now discusses this issue in adequate depth. Thank you!

Reviewer #4 Comment 3. In Figure 2b, there are some upper bands in lane 3. Please explain.

Our response: Thank you for your comment. We assume this is related to Figure 2C (not Figure 2b) which shows results of gel electrophoresis. We clarify that in the synthesis process we use exonuclease to cleave all the linear ssDNA, and the products are directly tested by gel electrophoresis. So, these cleaved fragments along with the enzyme and protein mixtures used in the synthesis are the potential cause of these blurry bands.

Reviewer #4 Comment 4. The kind and source are important in some CRISPR cases. In this work, EnGen® Lba Cas12a (Cpf1) protein from New England Biolab was used. What is the performance of DNACER with other kinds of cas12a (e.g. As Cas12a) or Lba Cas12a from other common-related reagent company?

Our response: In agreement with the reviewer's suggestion, we have tested the performance of AsCas12a, and we added this information in the Supplementary Figure S39. The reaction conditions

for AsCas12a were identical as for the LbCas12a assays. The corresponding information has also been added to Results, Section 10 in the integrated manuscript.

Reviewer #4 Comment 5. According to Figure 4b&c, the background of Cir-amplifier increases with the length of dsDNA and linker. Please explain the reason.

Our response: The old Figure 4B has been renumbered to new Figure 7A, and the old Figure 4C has been renumbered to new Figure 7B. In new Fig 7A, we evaluated the dsDNA length in Cir-reporters for the activation of a standard CRISPR/Cas12a biosensing system (with a 3nt ssDNA linker), the background for different dsDNA length in the Cir-reporter was consistent (shown in the figure at 0h), since their linker length is the same (3nt). In the new Figure 7B, we evaluated the effect of linker length from 0nt to 10nt in the Cir-reporter with the same dsDNA length of 18nt. Higher background was observed for longer linker length, since the FRET distance between the fluorophore and quencher was increased for these longer linker lengths, leading to lower FRET efficiency (reduced quenching effect and high background).

Reviewer #4 Comment 6. In Figure D6, the selectivity of DANCER should be studied.

Our response: In the integrated manuscript, the selectivity from typical sample interferences has been established by conducting the assay in relevant samples such as blood plasma (new clinical and previous animal data in Figure 9C and 9D and in human saliva (new data, Figure S44).

Reviewer #4 Comment 7. In Figure S5, were Cas12a RNP-2 and Cas12a RNP-1 designed to recognize different targets? It was understood that RNP-1 could only recognize the target DNA to achieve Cir-amplifier cleavage, while RNP-2 recognized the cleaved product "Linearized" to achieve "DNACER".

The Figure B/C showed that the Cas12a RNP1-RNP2 biosensing system significantly increased the signal of fluorescence response. However, the difference in signal intensity between "Two Cas12a RNP" and "One Cas12a RNP" in Figure d was not significant. Please optimize the experiment and add a detailed analysis.

Our response: Please note changed figure numbering in the now integrated manuscript. We confirm that, in Figure S42 (original Figure S5 in DANCER manuscript), Cas12a RNP-1 and Cas12a RNP-2 were designed to recognize different targets. Cas12a RNP-1 was designed to recognize the target DNA, and Cas12a RNP-2 was designed to recognize the re-linearized Cir-reporter. This is now been clarified in the combined manuscript at Results, section 10.

Regarding the second part of this comment that there is a conflict between what is shown in Figure S42 B/C and in Figure S42 D. We clarify that there is no conflict. This is because Figures S42 B/C are comparing the autocatalytic CRISPR/Cas system to a standard CRISPR/Cas – where autocatalysis ensures a significantly increased signal. In Figure S42 D both 1 RNP system and 2 RNP system are autocatalytic. In this case large differences are not expected – but the difference is still statistically significant.

To clarify the potential confusion we have now added the labels Autocar-2 and Autocar-3 to Figure S42.

Reviewer #4 Comment 8. In Figure D6a, the authors applied the Cir-amplifier substrate for the detection of ctDNA in non-diluted plasma samples. Does the complex detection environment affect the stability of the Cir-amplifier substrate in this experiment?

Our response: We now added these data in Figure S43 referred to from the main manuscript where we added the phrase “we confirmed the stability of the synthesized Cir-reporter in clinically relevant human plasma and saliva samples (over 1 hour at room temperature, Figure S43” in Results Section 11.

Reviewer #4 Comment 9. In the design, the authors used click chemistry to prepare Cir-ssDNA. What are the advantages of this method over the conventional T4 DNA ligase-based method? According to the literature, T4 ligase-catalyzed DNA cyclization method has high ligation efficiency and high purification efficiency with exonuclease treatment.

Our response: The advantages of the click chemistry method comparing with the T4 DNA ligase method are as follows:

1. Magnetic beads utilised in the click chemistry synthesis procedure improve purity of single ring ssDNA synthesis. In the click chemistry method, we used streptavidin-modified magnetic beads to immobilize the biotin labelled ssDNA, afterwards all the free ssDNA remaining in the solution has been removed simply by washing. Then, the click chemistry method was applied to link the head and tail of ssDNA to form the Cir-ssDNA. All the linear ssDNA molecules remaining after this step were removed from the sample using Exonuclease VII. Then, the final Cir-ssDNA product can be purified and concentrated from the beads. None of these options exist in the T4 DNA ligase method, which was found to form the intended circular structures as well as different size of rings/circles which were unintended.
2. High synthesis efficiency. The synthesis efficiency of click chemistry method was over 90% as shown in Figure S36C in the revised manuscript.
3. Feasibility of high concentration of Cir-ssDNA solution. In the click chemistry method, we used streptavidin modified magnetic beads to immobilize the biotin labelled ssDNA, which was applied to concentrate the synthesized Cir-ssDNA (Method 13). The concentration of such a small size circular DNA can be difficult after T4 ligase treatment.

To respond to this comment, the following correction has been made to the manuscript:

In the Method 13 we added a phrase “This method was introduced to improve purity of single ring synthesis and to improve product concentration. The use of magnetic beads makes it possible to concentrate Cir-ssDNA”.

Reviewer #4 Comment 10. In the manuscript, the authors indicated that "The Cir-amplifier was assembled by mixing Cir-ssDNA with fluorophore labelled cDNA (Texas-cDNA-BHQ2)." What is the ratio of added Cir-ssDNA to Texas-cDNA-BHQ2?

Our response: The ratio of added Cir-ssDNA to Texas-cDNA-BHQ2 was 1:1. This ratio was added to the phrase “at a ratio of 1:1” in Method 14.

Reviewer #4 Comment 11. In Figure 4, were the "L-dsDNA" in Figure b and the "Linearized Cir-amplifier" in Figure e the same structure? It was not clear in the manuscript.

Our response: L-dsDNA was formed using ssDNA with its cDNA, while the linearized Cir-amplifier originates from the Cir-reporter, which was cleaved by activated Cas12a RNP. (Note that Cir-amplifier is now called Cir-reporter in the combined manuscript).

To address this comment, in the revised manuscript we added a clarification into the caption of new Figure 7A which now says “L-dsDNA has been formed by using ssDNA with its c-DNA”.

Reviewer #5

Major concerns:

Reviewer #5 Comment 1. The authors should provide more information to improve the conformation of Cir-ssDNA. Current evidences were based on electrophoresis (Fig. 2b) and reference 12 (Cir-ssDNA containing a single triazole, amide or phosphoramidate backbone linkage), and the two Cir-ssDNAs are different.

Our response: We understand that the reviewer seeks additional evidence (beyond Fig 2b, now renumbered as Figure S36B that Cir-ssDNA is indeed circular.

In this study, we verified the formation of Cir-ssDNA using three different methods. The first method is electrophoresis (previous Figure 2b, now Figure S36B), which confirms the difference between Cir-ssDNA and linear ssDNA. The second method is to investigate the integrity of the Cir-mediator using exonuclease III (now added as Figure. S10B). In comparison to linear ssDNA which can be completely degraded by exonuclease III due to the exposed free 3' terminal, this Figure shows that the synthesized circular ssDNA survives exposure to the endonuclease. This proves the formation of a circular structure because closing of the circle prevents exonuclease access to the 3' terminal. The third approach is to use the FRET method to demonstrate the formation of Cir-ssDNA (Figure 6B). The Cir-reporter was formed by Cir-ssDNA and its corresponding fluorescent cDNA (5'-Texas Red; 3'-BHQ2). The Linearized Cir-reporter (L-Cir-reporter) was formed by linear ssDNA and its corresponding fluorescent cDNA (5'-Texas Red; 3'-BHQ2). As shown in Figure 6B, the fluorescent signal of L-Cir-reporter is 16.5 times higher than that of Cir-reporter, which further confirms the formation of Cir-ssDNA.

Reviewer #5 Comment 2. Apart from initial circle conformation, the author optimized lengths of the main part DsDNA and linker ssDNA to reduce the background signal using a testing system, what is the background in the real sample? The author should explain why longer sequence of the DsDNA or linker ssDNA gives rise to higher background?

Our response: The background of Cir-reporters (previously called Cir-amplifiers) in a plasma sample is shown in Figure 9A.

We clarify that longer linker ssDNA sequences give rise to a higher background because the fluorophore and quencher in Cir-reporters are separated by the linker distance. The fluorophore quenching efficiency is due to FRET effect which decreases at longer distances.

To address this comment, in the revised manuscript we added a phrase to the caption of Figure 7B saying “Longer linker lengths produce higher background due to reduced FRET effect with more distant quenchers.”.

Longer dsDNA sequences (at constant linker length) give rise to a higher background compared to shorter sequences because the main effect reported in this paper, restricted Cas12a activation for circular targets depends on how highly tensioned these structures are. Longer dsDNA sequences produce circular structures that are under less tension and better able to unwind the two strands and achieve a spatial configuration needed to bind to guide RNA. This is extensively covered in the discussion section of the revised manuscript.

Reviewer #5 Comment 3. Many CRISPR-based amplification-free methods have been reported (Front. Microbiol. 2021, 12:751408), also with different kind of optimization. The authors claim that they achieved the signal increments of 16.5 or 20 times, why not compare the values with the reported ones? and what is detection limit in real sample in addition to a model one (1 aM).

Technique issues:

Our response: Our detection limit in plasma is 1 aM, as shown in Fig. 9A.

The only method with a comparable LOD to ours (5 aM, but 240 min detection time) (Shi, K. et al, A CRISPR-Cas autocatalysis-driven feedback amplification network for supersensitive DNA diagnostics. Science Advances 7, 2021) has been carefully compared with ours in the discussion of the revised manuscript. This work not only shows a longer detection time, but it has a range of other vulnerabilities. The remaining methods do not show comparable performance, see for example a recent review paper doi: 10.3389/fmicb.2021.751408, Table 1 which lists alternative amplification free methods and their performance.

Technique issues:

Reviewer #5 Comment 1. The (DANCER) scheme is not clear. It is necessary to clearly describe what each element represents.

Our response: All schemes in the revised integrated manuscript have been redrawn and the graphical elements have been fully explained (see for example Figure 4 where they are captioned).

Reviewer #5 Comment 2. Normally, the target DNA pairing with the crRNA should have a length of around 20 nt. The authors claimed that the best one was the 18-nt Cir-amplifier. This means it only

has 15 nt to pair with the crRNA (i.e., TTT is the PAM region), which should compromise the detection sensitivity.

Our response: We are aware of this compromise, but we needed to balance the background reduction with Cas12a activation. To this aim, we decreased the length of dsDNA in Cir-reporters to reduce the background signal (Figure 7A in the integrated manuscript), without significantly reducing the Cas12a activation efficiency of dsDNA. As shown in the new Figure 7A, the Cas12a activation efficiency of 18nt dsDNA is around 50% of that of 25nt dsDNA.

Reviewer #5 comment 3. The authors demonstrated an application based on the lateral flow strip. It seems the Control group also showed two bands in the strip. Why?

Our response: The signal on the test line in the control group may be due to some non-specific binding of the FAM Ab from the conjugation pad (FAM Ab-Au NP) to the anti-FAM Abs on the test line. In addition, residual activation of *trans*-cleavage by the circular DNA structure (observed as the background signal in fluorescence-based tests) also contributed to this background visible on the test line.

To address this comment, we added in the caption of Figure 9F: “The test line in the CRC LFA (left) shows a darker colour compared to the control LFA (right). The background signal in the control LFA is attributed to nonspecific binding or residual Cas12a activation”.

Reviewer #5 Comment 4. There is a concern about the competition between the crRNA and the cDNA to bind with the Cir-ssDNA even when there is no target. If this is not an issue, the author might have to discuss it and explain why.

Our response: As explained in Method 14, during the preparation stage we first mix the Cas12 protein with crRNA (1:1 molar ratio) to form the Cas12a RNP. Separately, we mix the Cir-ssDNA with its cDNA (1:1 molar ratio) to form the Cir-reporter. Afterwards, the two solutions are mixed together to form the reaction mixture. Therefore, there is no competition.

Reviewer #5 Comment 5. More tests should be performed to use clinical samples to demonstrate the accuracy of this method.

Our response: In the revised manuscript, we added clinical data in cancer patients shown in Figure 9D. The added manuscript text reads: “Finally, we tested the ability of AutoCAR-3 to detect PI3KCA H1047R mutations directly from human plasma. Blood samples from patients with tumors known to harbor PI3KCA mutations, as well as control cancer patients without that mutation, were selected from the Molecular Screening and Therapeutics (MoST) program (ACTRN12616000908437). Plasma samples were assessed for mutation detection using the same PIK3CA H1047R biosensor method (test tube) as for the CRC mice, with a 2-hour incubation at ambient temperature. The results show statistically significant discrimination between cancer and control samples (Figure 9D, S44), which can also serve as a prove for the feasibility of DNA single nucleotide polymorphism detection. In addition, we tested if our system can be used for the detection of target DNA in other types of clinical samples such as saliva, by spiking of 1 fM PIK3CA H1047R mutation sequence. The results

also shown that the presence of targeted PIK3CA H1047R mutation sequence was detected with 100% accuracy (Figure S45)".

Responses to the comments for the manuscript "Topological barrier to Cas12a activation by circular DNA nanostructures facilitates autocatalysis" (manuscript NCOMMS-23-12500-T, now under revision)

Manuscript corrections related to NCOMMS-23-12500-T have all been transcribed to the integrated manuscript, and are marked by **blue** highlights.

Reviewer #1

Reviewer #1 comment 1. The statement in the abstract, "increase the achievable trans-cleavage rates per target by over 3 orders of magnitude," is misleading because this paper used the trans-cleavage of Cas12a RNP to activate more RNPs that generate amplified signals, and not to significantly alter the trans-cleavage rate of Cas12a.

Our response: We acknowledge that this might be confusing. In the revised manuscript we modified this and similar statements elsewhere as to "increases the achievable reporter cleavage rates per target"

Reviewer #1 comment 2. Previous literature has revealed that Cas12a trans-cleaves ssDNA into small fragments with lengths of 2 or 4 nt (Cell Res, 2018, 28, 491). Since the cir-mediator used in this paper contains only a 2 nt single-stranded region, direct evidence is needed to demonstrate whether it can be trans-cleaved by an activated Cas12a RNPs.

Our response: The evidence for cleavage of the 2nt region of Cir-mediators is now shown in Figure 6. in the revised manuscript. In this figure we modified the 3' and 5' ends of cDNA in the Cir-mediator with a fluorophore and a quencher forming Cir-reporters. The fluorophore and quencher are separated by a single-strand linker region. In the circular conformation, FRET effect ensures that the fluorophore is well quenched by the proximal quencher. Upon exposure to activated Cas12a, a significantly increased fluorescent signal indicates a successful cleavage of the single-strand region, as the fluorophore quenching ceases upon linearisation

Reviewer #1 comment 3. The Cir-mediator used in this paper (Fig 1B) is cleaved to form an 18 bp dsDNA, but only a 14 nt sequence can hybridize with gRNA spacer. However, the results in Fig S3 showed that the 18bp dsDNA induced only a very weak signal, slightly above the negative control. Therefore, it is difficult to imagine how an effective chain activation could be generated using the cir-mediator.

Our response. Please note that figure numbering in the revised manuscript is now modified. We clarify that this comment is due to our imperfect labelling of nanostructures in Figure S3 which we have now corrected. The correct structure corresponding to that shown in Fig 1B is 20nt/18bp. The original label was 20 nt/bp, which is not sufficiently precise to reveal its structure. 20 nt is the overall target strand length, while 18 bp is the dsDNA region after it has been combined with its cDNA. After this correction of labels, we can see that the signal levels for of 20nt/18bp for both ssDNA and dsDNA are significantly higher than Neg- control. The corresponding changes has been made in Figure S3.

Although the overall signal level is much lower than for the total length of 24nt or 27nt, even this compromised activation efficiency is sufficient to set up an autocatalytic chain reaction.

A chain reaction is successfully established even if there is a very small effective increase above unity in the number of Cas proteins activated in a single autocatalytic cycle of target/fake target detection. The mathematical model of this is simple and its results well known e.g. from the Covid pandemics where the key number was the basic virus reproduction rate R_0 . The value of R_0 which were even slightly higher than unity means that the epidemics was growing. While the details of our rate equation model are more complex than virus reproduction, the core idea is the same.

Reviewer #1 comment 4. The concentrations of trans-cleaved reporters in Fig. 4 D-F are significantly higher than those of the reporters used in the experiments. As an example, in Fig. 4E, the maximum concentration of ssDNA reporter is 5 μM , while the concentration of cleaved reporters can reach 20 μM according to the y-axis. Why is there such a huge difference?

Our response: We apologise for this error which we have now corrected. The correct labelling of the y-axis of Figures 4D to 4F is now added. (The figure numbers are currently modified).

Reviewer #1 comment 5. The blank signal in Fig. 5 exhibits a significant variation between 0.8 and 4, which requires a comprehensive discussion. Moreover, the marker curve indicates an increase in signal with increasing concentration, but the actual samples saturate at approximately 10 copies/ μL , and the signal decreases even at higher concentrations. This phenomenon needs to be explained.

Our response: Several factors can potentially contribute to the signal variation in Figure 5 between 0.8 and 4. Firstly, there is a general background signal differences in AutoCAR system in response to different types of targets, such as genomic DNA (Figure 5B) or genomic RNA (Figure 5D). This is because genomic nucleic acid materials lead to potential off-target activation of Cas12a RNP, a well-known effect in CRISPR/Cas gene editing (<https://doi.org/10.1021/acs.analchem.1c05499>). This additional background which may be generated by genomic nucleic acid materials can be the cause for the phenomenon of signal saturation at ~ 10 copies/ μL . To respond to the Reviewer's comment, additional explanation has now been added into the caption of Figure 5 which reads "Due to potential Cas12a off-target effect in genomic nucleic acid materials, comparison and screening of different Cas12a targeted sequences can potentially improve the overall background and performance". In addition, batch-to-batch differences between synthesized Cir-mediators may also contribute to the variation of background signal. Also, we emphasise that, the fluorescence signal intensity in this figure is given in arbitrary units ([a.u.]). The absolute value of a reading in arbitrary units may not directly represent the difference of reactions, but the relative changes within the same reaction conditions does reflect the difference between tested samples.

Reviewer #1 comment 6. It is important to clearly indicate whether synthetic genome fragments or real pathogenic genomic DNA/RNA were used during ultra-sensitive DNA and RNA detection. Furthermore, the practical application of the method should be demonstrated using real samples in excess of 50.

Our response: The captions and the referring text to Figures 5B and 5D now clearly state that these figures are for whole genome DNA/RNA.

To address the comment about practical application of the method we now added new figure panels showing assay results for 50 clinical samples (in blood plasma, added in Figure 9D, and also in saliva in Figure S44).

Reviewer #1 comment 7. In the AutoCAR system, the amount of activated Cas12a RNPs increases continuously. It is challenging to accurately determine the Michaelis-Menten constant in this system due to its complex dynamics.

Our response: The dynamics is indeed complex, and this is why we created our rate equation model, to be able to fully capture and explain this complexity, as well as in order to be able to make predictions. (See Supplementary Information, Supplementary Note). Our model allows to establish, in the Briggs-Haldane approximation, that the observed reporter cleavage rate (in specific reaction condition) is proportional to the concentration of activated RNPs (Equation 24). While the concentration of activated RNPs vary with time, this variation is slow (over minutes) compared with the catalytic rate (*trans*-cleavage rate) of several cleavage events per second. This difference of time scales supports the application of the Michaelis-Menten theory in defined time windows over which the activated RNP concentration does not vary significantly. This is of course an estimate only.

To address this comment, we added the following more explicit statement in the Supplementary Note “This model is limited to the case of molecular concentration transients that are much slower than the *trans*-cleavage rate” in the Supplementary Note.

Reviewer #2

Reviewer #2 comment 1: The model and experimental data support the fact that the approximate number of activated Cas molecules per single target molecule (R cas/target) has increased by three orders of magnitude in the AutoCar system (Table AS2, reaction time of 1500 s). However, this seems inconsistent with the reported aM detection limit reported in figure A5 for DNA and RNA detection. Indeed, aM detection limits mean approximately six orders of magnitude lower than the well-established LoD reported for pre-amplification free Cas12a detection systems (1-10 pM). How is it possible to achieve six orders of magnitude lower LoD with a three orders of magnitude increase in activated Cas12 enzyme per target? The authors should critically analyze and deeply discuss this point in the main text since it cannot be simply ascribed to the longer time of analysis (1-1.5 h, see methods 4.12 and 4.13) that is longer compared to the time of the assay reported for the generation of the model (1500 s, 25 minutes).

Our response: We clarify that three orders of magnitude increase (approximately a factor of a 1000) in activated Cas12a enzyme per target has been achieved after 25 minutes. We remind that the number of activated Cas12a proteins depends exponentially on the reaction time, as explained in our model

and generally common in autocatalysis. In particular, this means that if we consider each of the activated Cas12a proteins found in the reaction mixture after 25 minutes, each of these proteins will create the exact same avalanche of new activated Cas12a proteins as the initial Cas12a molecule activated by the original target (as long as the reaction substrates, inactive Cas12a proteins and Cir-mediators remain in excess). This means that after ~ 1 h (approximately 2 x 25 minutes) we will observe the increase in the number of activated Cas12a proteins per target by a factor of 1000x1000 which is precisely six orders of magnitude the Reviewer is asking about.

To clarify this point in the revised manuscript we added the following comment in the caption to Figure 5 where 1 aM LOD first appears. “The 1 aM LOD is consistent with exponential growth in the number of Cas12a proteins activated by a single target (~ 3 orders of magnitude in ~25 minutes, over approximately twice the duration (1 hour) yields the LOD increase of ~ 3 orders of magnitude squared, so 6 orders of magnitude lower compared with well-established LOD values for pre-amplification free Cas12a detection systems (1-10 pM).”

Reviewer #2 comment 2: In addition, the authors should add two experimental panels (one for DNA and one for RNA detection) in figure 5 reporting the raw kinetic profiles (fluorescence vs. time) of the trans-cleavage activity - with appropriate control experiments - in the range of concentrations (from aM to fM) to clearly demonstrate the reported sensitivity.

Our response: To respond to the reviewer's comment, we now added a supplementary figure (Figure S34) containing the requested data.

Reviewer #2 comment 3: Ensemble sensors rely on a "large" sensing volume, leading to a response change that is the aggregate response from the sensor. To achieve ultra-low concentration sensitivity, as has been clearly reported recently (see ACS Sens. 2023, 8, 3, 941–942), sampling issues are important for concentrations below fM, where the typical sample volumes of μL do not ensure an ample supply of analyte. Hence, the authors should highlight if the experimental procedures here reported meets this requirement.

Our response: The root cause of sampling issues at very low concentrations and volumes referred to by this Reviewer is the statistical (Poissonian) distribution of molecules. For example, for 1 molecule per 1 μL sample the probability of finding no molecules at all in that volume is approximately 24%. This statistical distribution leads to inaccuracies in the measured number of molecules. To address this comment, we directly calculated how accurate is our procedure of measuring all quantities in triplicate and we added the clarification as the new Statistical Methods section in the Supplementary Material as Method S19. The added section reads:

“Our experimental procedure of measuring molecular concentrations and related quantities in triplicate is not significantly affected by sampling issues at molecular concentrations and sample volumes used in this study, despite their low values. All our measurements are taken in triplicates (X_1 , X_2 , X_3) and then we use the average value of readings X . If X represents the experimentally obtained number of molecules (the “sample estimate” in the language of statistics), it is possible to calculate the probability that this X represents the real average number of molecules, m , in these samples with 20% accuracy. To this aim we need to calculate the probability $P(0.8 X < m < 1.2 X)$. This probability is the same as the probability $P(10/12 m < X < 10/8 m) = P(10/4 m < X_1 + X_2 + X_3 < 30/8 m)$.

The random variable $X1+X2+X3$ has a Poissonian distribution with the constant $\lambda= 3m$.

Therefore $P= \sum_{k=0}^{\frac{30m}{8}} \frac{(3m)^k}{k!} e^{-3m}$, and this expression makes it possible to calculate the numerical value of probability P .

For 1 aM samples at 10 μ L volume used in this work for plasma measurements the value of $m=10$, $P=0.75$. For 10 aM samples at 10 μ L volume, the value of $m=100$, and $P=0.998$. For larger volumes and higher concentrations P is higher than 0.998. All these probabilities are close to unity which means that our sample estimate X (the average of $X1$, $X2$, $X3$) closely represents the value of m .”

The calculation can be easily modified to describe measurement accuracy different than 20%.

Of course, the accuracy of measurements is also affected by standard experimental factors such as pipetting accuracies, evaporation etc, but this is typically addressed in the literature by taking measurements in triplicate.

Reviewer #2 comment 4: The study lacks specificity tests, which is a significant limitation. The authors should characterize the AutoCar system with single and double MM DNA targets to determine if the platform is specific to the target sequence.

Our response: To address this comment, we added a new Figure S40 referred from Results 10 in the main text as indicated earlier. The added text reads “The detection of single nucleotide mismatches in the target sequence has also been demonstrated (Figure S40)”. Please note that our clinical data in animal cancer samples (Figure 9C) and newly added clinical data for human plasma samples (Figure 9D) also confirm SNP accuracy, because the oncogenic mutation PIK3CA H1047R has a single SNP difference with wildtype (Zhou et al. 2022. *Chemical Communications*, 58(28), 4508-4511; Thierry et al. 2010. *Nucleic acids research*, 38(18), 6159-6175). While we provided new data only for single MM DNA, it is well established in the literature that double MM detection is easier than single MM detection (Long T. Ngugen et al. 2020. *Nature Communications*, 11:4906; Gootenberg et al. 2017. *Science*, 356(6336), 438-442.).

Reviewer #2 comment 5: The study appears to be incomplete in terms of statistical analysis. Although many histograms are presented, the associated data points and number of replicates should be reported. Image captions should include the number of replicates performed in each experiment, and when P values are defined, the authors should specify how they were calculated.

Our response: In the revised manuscript we added a new Statistical Methods section (Method S19 in the Supplementary Information) which states the details of statistical analysis and the number of replicates ($n=3$). We understand that the “histograms” are meant to be “bar charts”. The P values are calculated by using two-tailed student T-test, they are adopted as conventional boundaries (e.g. $P<0.05$ is usually taken as the significance level allowing to reject a null hypothesis that the two variables have the same distribution). The number of replicate number and the P value information has now been presented in the manuscript as ($n=3$), (* $P<0.05$, ** $P<0.005$, *** $P<0.001$).

Reviewer #2 minor comment 1: In most of the assays, a concentration of 50 nM Cir-mediator is used, but in Figure A4 panel F, this concentration is not investigated. The authors should add this data to the panel, and the raw fluorescence data associated with panel F should be included in the supporting information.

Our response: Higher concentrations of circular constructs, up to 200 nM were used in Figure 8B, and they also show exponential trends. In Figure 4F (now Figure 4 H, I, J) the concentration of 20 nM is already showing an exponential increase of the total cleavage rate, which means that the circular constructs were in sufficient excess for autocatalysis. In many assays we used 50 nM Cir-mediator more tends to make sure the reaction will not run out of functional Cir-mediators.

Reviewer #2 minor comment 2: Figure 2B shows fluorescence vs ssDNA length, not the trans-cleavage rate, so the text should be revised accordingly.

Our response: This is revised in the integrated manuscript in the caption of Figure 2B.

Reviewer #2 minor comment 3: The colors of the lines and the legend in Figure S6 are not very clear.

Our response: This is revised in the integrated manuscript in Figure S6.

Reviewer #2 minor comment 4: In Figure 2G, the authors should explain the drift observed in the gel electrophoresis of dsDNA sequences. They should also demonstrate that Cir-mediator is not cleaved over time using gel electrophoresis, not only the "linear" dsDNA sequence as reported.

Our response: The drift of the dsDNA bands in the gel may be due to the uneven gel thickness. Based on the reviewer's comments, we also added a new gel figure for pre-activated Cas12a -treated Cir-mediator (Figure S10A), which indicate the formed Cir-mediator structure is not completely cleaved by pre-activated Cas12a *trans*-cleavage.

Reviewer #2 minor comment 5: Section 7 reports that the experimental method used to analyze the *H. pylori glm* gene fragment is 4.11, but the image caption indicates that it is method 4.12. The authors should check all the text and image captions to ensure consistency regarding the methods mentioned.

Our response: Thank you for your comment. The correct method now is Method 10 in the integrated manuscript. We have also carefully checked the entire submission and we believe there are no similar errors.

Reviewer #2 minor comment 6. The caption of Figure S6 does not include information about the length of the ssDNA sequence, its concentration, or other experimental conditions. The authors should improve the clarity of their experimental conditions.

Our response: This additional information was added to Method S2, which has been used in Figure S6.

Reviewer #2 minor comment 7 For the data in Figure S17, it is not clear whether a target activator of the ssDNA or dsDNA was used or at what concentrations. The authors should clarify these experimental details.

Our response: The required information is now added to Figure S19 in the integrated manuscript (note a new number of that figure in the revised manuscript).

Reviewer #3

Reviewer #3 comment 1. The authors should show the cleaved product band in the ssDNA gel and its absence in the dsDNA gel, or else quantify them (Fig. 2G).

Our response: Trans-cleavage will cut the ssDNA into fragments shorter than 2nt, which will not be able to form a band in the gel. Instead, to address this comment we quantified band intensity changes using image analysis and Image J. While this is not a precise method, we added this information to the manuscript in Figure S9.

Reviewers' Comments:

Reviewer #1:

Remarks to the Author:

In this version, the authors have addressed all the queries, and acceptance is recommended.

Reviewer #2:

Remarks to the Author:

The authors conducted numerous additional experiments and implemented significant changes in the text to address all the comments that arose. I believe the article is now ready for acceptance without the need for further modifications.

Reviewer #3:

Remarks to the Author:

Accept

The authors originally submitted two manuscripts for back-to-back publication. During the review process, it appeared that a single comprehensive article would have been more appropriate.

With this revision, the authors did an excellent job of combining the two papers into one. Moreover, the authors addressed the concerns and provided a solid statistical analysis. I think that the paper and the technology involve an ingenious approach and a viable design, and hence, the publication can move forward.

Reviewer #4:

Remarks to the Author:

The authors have addressed all my concerns and improved the manuscript significantly. I think it can be accepted.

Reviewer #5:

Remarks to the Author:

The authors have addressed my comments properly and I am happy with their responds and also the editor's suggestion to merge the two manuscripts. I have no comment to add on and recommend to accept it for publication.